# ReCAPA: Hierarchical Predictive Correction to Mitigate Cascading Failures

**Xiyin Zeng**[1,2*], **Yuyu Sun**[2*], **Haoyang Li**[2*], **Shouqiang Liu**[2], **Hao Wang**[1†]
[1]Hong Kong University of Science and Technology (Guangzhou)
[2]South China Normal University
[*]Equal contribution      [†]Corresponding author

## ABSTRACT

Vision–Language–Action (VLA) systems follow instructions to execute multi-step tasks in multimodal environments. Recent VLA approaches typically rely on post-hoc correction mechanisms or operate under fixed task decompositions and alignment schemes. However, once an intermediate step is mis-specified, local errors propagate through subsequent steps and eventually accumulate into cascading failures. To mitigate this compounding effect, we propose Predictive Alignment and Planning Architecture (ReCAPA), a framework that uses prediction and contrast to adjust deviations across three levels: actions, subgoals, and trajectories. Semantic alignment is enforced at all levels using a Sinkhorn-based module and a Score-field module. The predictive correction and alignment, jointly updates the action-generator in the training phase, enabling it to adjust fine-grained steps to remain aligned with the overall intent. We further introduce two new metrics to quantify error propagation and recovery processes in tasks, capturing how mistakes spread and fade over long-horizon execution. Experiments show that ReCAPA achieves competitive results on embodied agent benchmarks such as VisualAgentBench, MineDojo, and AI2-THOR, outperforming strong proprietary and open-source Large Language Model (LLM) baselines. Project page: https://sunandreas0437-svg.github.io/recapa-project-page/

## 1 INTRODUCTION

VLA agents powered by LLMs are increasingly applied to long-horizon tasks in embodied environments, such as household manipulation, indoor navigation, and multi-turn human-robot dialogue (Jiang et al., 2023). These tasks require perception, planning, and grounded execution under natural language guidance. Yet many systems struggle to generalize across multi-step environments, as they lack structured visual–language grounding and often collapse under semantic drift and cascading errors(Comanici et al., 2025; Anthropic, 2024a; Achiam et al., 2023).

Recent VLA agents such as Re-ReST (Dou et al., 2024), LLaMAR (Nayak et al., 2024), and City-NavAgent (Zhang et al., 2025a) shift from selecting actions step by step to executing subplans, which decompose the overall plan into executable segments. Both TrajPrompt (Tsao et al., 2024) and PRET(Lu et al., 2024) incorporate alignment between instructions and trajectories through semantic matching, thereby strengthening execution coherence with the task intent. However, semantic drift and error propagation remain key bottlenecks of long-horizon reasoning. On one hand, post-hoc correction and static execution makes it difficult to flexibly adjust its actions when execution deviates. On the other hand, relying only on local alignment leads to each step being optimized in isolation, thereby drifting from the overall intent easily. In benchmarks such as VirtualHome(Puig et al., 2018) and AI2-THOR(Kolve et al., 2017), even a single subgoal error can degrade the performance of subsequent steps by over 60% (Zhong et al., 2024; Zhu et al., 2021). In sum, current agents may encounter execution–goal divergence and cumulative rollout errors.

Action errors may compound rapidly in the short term, whereas strategy misalignments unfold more slowly distort the overall plan. Yet many methods reflect only at a single level, inevitably leaving other level propagation unchecked (Sun et al., 2023) (Zhou et al., 2024). To address this, we refine actions and subplans during training using higher-level supervision. This supervision enforces align-

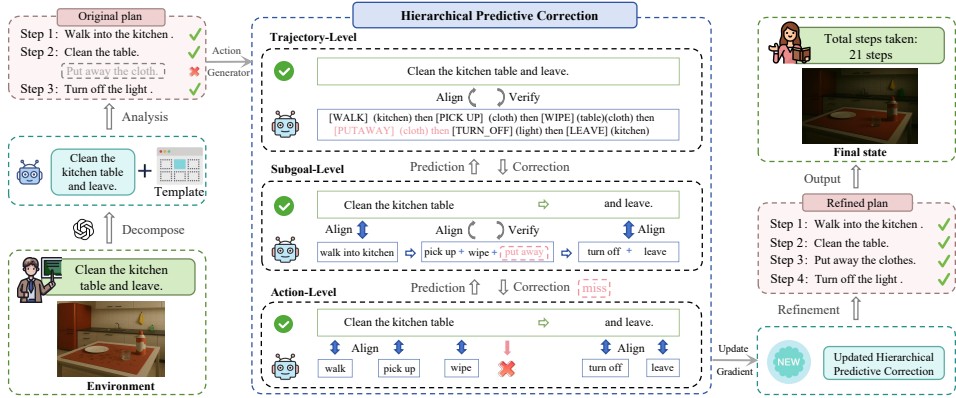

Figure 1: Overview of the ReCAPA framework. The LLM first generates the agent's original plan through task decomposition. Hierarchical Predictive Correction module executes the task and produces fine-grained corrective signals to guide execution updates.

ment across different levels of planning. During inference, the action generator can detect potential deviations early. It then favors behavior compositions that better match the overall task intent.

Guided by the above considerations, we propose ReCAPA as shown in Figure 1. ReCAPA seperates trajectories into action-, subgoal-, and trajectory-levels. Unlike prior methods that rely on fixed decomposition and apply only after failures, ReCAPA introduces Hierarchical Predictive Correction (HPCC) with cross-level alignment signals to prevent errors from compounding. These alignment signals are obtained by comparing trajectory embeddings with the prompt embeddings, using a Sinkhorn-based (Cuturi, 2013) module and a Score-fieldSong et al. (2020) module.The Sinkhorn-based module aligns the overall trajectory distribution with the prompt, thereby guiding execution to remain consistent with task intent at the trajectory level. In parallel, the Score-field module provides step-specific alignment across the remaining two levels. At the subgoal level, HPCC predicts the trajectory-level representation and evaluates its consistency with realized trajectory rollouts, updating predictions when mismatches are detected. At the action-level, ReCAPA predicts the subgoal representation and revise fine-grained execution errors. Together, HPCC and prompt-trajectory alignment enable early, cross-level corrections that reduce drift.

To properly assess these benefits, evaluation should go beyond success rate (SR) to also capture how errors propagate, accumulate, and dissipate throughout execution, which existing benchmarks largely overlook. To fill this gap, we introduce two diagnostic metrics in long-horizon reasoning: Error Propagation Rate (EPR) quantifies how mistakes compound across steps, and Propagation Attenuation Coefficient (PAC) measures how errors attenuate or dissipate over time. Together, these metrics capture how failures both spread and decay, providing diagnostic tools to evaluate ReCAPA's stability. Our contributions demonstrate the following advancements:

- We propose ReCAPA, a framework operationalizes hierarchical correction by coupling multi-level predictive representations with prompt–trajectory distributional alignment, allowing deviations to be anticipated and corrected earlier.

- We introduce two diagnostic metrics for error propagation in long-horizon reasoning: EPR quantifies the propagation of errors across future steps, while PAC captures the system's ability to recover by measuring how quickly error impacts dissipate over time.

- ReCAPA outperforms strong LMM baselines in terms of success rate, achieving +5.65% on VisualAgentBench, +9% on MineDojo, and +7% on AI2-THOR.

## 2 RELATED WORKS

**Decomposition Approaches** Task decomposition has been explored in methods such as HIRO (Nachum et al., 2018) which executes fixed interval subgoals and EPO (Zhao et al., 2024) which applies reward modeling for hierarchical planning. LLaMAR (Nayak et al., 2024) and CityNavAgent

(Zhang et al., 2025a) both follow pre-defined multi-stage subgoal pipelines, but they introduce novel modifications to improve task execution success. LLaMAR improves upon multi-stage task decomposition and policy optimization, while CityNavAgent decomposes subgoals and uses memory of past trajectories to aid planning. However, these static approaches may struggle to adapt when the initial decomposition is flawed, leading to errors in dynamic environments.

**Error-Correction Mechanisms** Addressing cascading failures across different temporal scales is critical Zhang et al. (2025b); Lan et al. (2024); Zheng et al. (2026). To address static planning, later works introduced feedback-based corrections. ReAct (Yao et al., 2023a), Reflexion (Shinn et al., 2023), and WALL-E (Zhou et al., 2024) provide step-level or episodic updates. AdaPlanner (Sun et al., 2023) revises subplans based on feedback to adapt to changing environments, while R3V (Cheng et al., 2024) self-reflects by generating and selecting multiple candidate paths. These distinct propagation patterns suggest that effective correction should span multiple modules, requiring consistent alignment across steps and overall goal Chai et al. (2025). Although these methods incorporate dynamic error-correction mechanisms and can adapt through feedback from the environment or their own internal processes, they still struggle to maintain consistency across different stages of task execution.

**Integration Attempts** To maintain consistency during executions, recent research has explored frameworks that integrate decomposition with semantic alignment. For example, TrajPrompt (Tsao et al., 2024) represents trajectories as segmented prompts and aligns them within a shared vision–language semantic space, thereby jointly embedding trajectory dynamics with scene semantics. HiP (Ajay et al., 2023) first decomposes long-horizon tasks into symbolic subgoals using a language model, and then aligns these subgoals with visual and action semantics to ground them into executable actions. VistaWise Fu et al. (2025) uses structured retrieval and external knowledge bases to enhance semantic grounding and task understanding Fu et al. (2025). However, these methods primarily focus on aligning substeps, which can result in correct subgoals but failed actions, making it difficult to maintain consistent alignment between overall intent and operations. In contrast, ReCAPA enforces cross-level predictive: lower levels forecast higher-level representations, and deviations trigger top-down corrections that use alignment signals to pull local decisions back to global goals and mitigate error propagation early.

## 3 METHODOLOGY

### 3.1 FRAMEWORK OVERVIEW

Most existing methods rely on pre-defined segmentation or post-hoc correction, making it difficult to anticipate errors and suppress their propagation, which often leads to failures. In addition, local alignment is often insufficient, as it lacks global feedback to correct misordered segments or handle ambiguous cases. To overcome this, we introduce prediction to expose deviations early. In the training phase, ReCAPA takes trajectory segments, prompt embeddings, and visual observations as input, with Hierarchical Predictive Correction (HPCC) representing trajectories at three levels. Prompt–trajectory alignment is computed at each level by comparing trajectory representations with prompt embeddings, producing cross-level consistency signals. These signals define the training losses, which are backpropagated through the predictors and encoders to update action and object selection.

At inference, ReCAPA uses environmental observations, the prompt, and historical trajectories as inputs. The execution network generates trajectories, while the LLM (GPT-4o-mini) provides task decompositions and completion markers. The three-tier correction mechanism refines the trajectory by resampling actions, adjusting subtasks, and using Sinkhorn for prompt alignment.

### 3.2 HIERARCHICAL PREDICTIVE CORRECTION

At the core of ReCAPA, HPCC predicts higher-level semantics from lower-level steps and reflects back corrective signals, supporting consistent task representations. HPCC organizes alignment into three levels: actions, which capture how fine-grained steps compose into short-term subgoals (e.g., [GRAB], [WALK], [WIPE] → cleaning); subgoals, which forecast trajectory outcomes and enforce

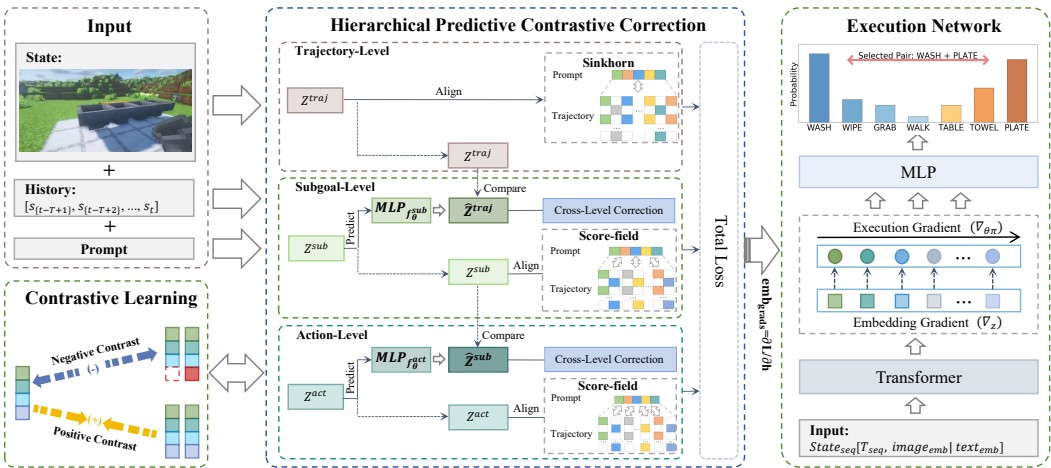

Figure 2: Overview of the ReCAPA training process. In ReCAPA, state, history, and prompt are encoded into hierarchical representations, with predictive and alignment losses guiding the action-generator to produce and revise actions.

causal order (e.g., washing before drying); and trajectories, which encode the task's overall intent and outcome.

HPCC takes the trajectory as input and turns it into multi-level representations at the action, sub-goal, and trajectory scales. Fine-grained actions combine to form higher-level subgoals, which together structure the task and encode sequential behavior patterns to anticipate overall task semantics. Specifically, at each level $l \in \{\text{action}, \text{subgoal}\}$, the model predicts the representation at level $l + 1$, based on a segment collection $\mathcal{T}^l$ constructed via sliding windows along the trajectory. The collection $\mathcal{T}^l$ at level $l$, optionally concatenated with visual embeddings from environment observations, is encoded into $\mathbf{z}^l$. We denote $\mathbf{z}^l \in \mathbb{R}^d$ as the vector-valued state representation at level $l$. The representation $\mathbf{z}^l$ is subsequently processed by a Transformer-based predictor, producing a predicted higher-level representation $\hat{\mathbf{z}}^{l+1}$. The representation $\mathbf{z}^l$ is subsequently processed by a Transformer-based predictor, producing a predicted higher-level representation $\hat{\mathbf{z}}^{l+1}$. Subsequently, the predicted representation $\hat{\mathbf{z}}^{l+1}$ is compared with the target representation $\mathbf{z}^{l+1}$ to construct cross-level alignment losses that regularize the lower-level representation $\mathbf{z}^l$.

The discrepancy between the cross-level predictor's estimate $\hat{\mathbf{z}}^{l+1}$ and the target embedding $\mathbf{z}^{l+1}$ is quantified by a prediction loss. The magnitude determines the strength of the supervisory signal from the level-$(l + 1)$ encoder that regularizes the lower-level representation $\mathbf{z}^l$.

Concretely, the cross-level contrastive loss $L_{\text{pred}}^l$ is implemented as an InfoNCE objective on $(\hat{\mathbf{z}}^{l+1}, \mathbf{z}^{l+1})$. $L_{\text{pred}}^l$ encourages the predicted representation to stay close to the intended higher-level signal while distinguishing it from distractors. During optimization, gradients are backpropagated through the predictor and the level-$l$ encoder, while the level-$(l + 1)$ target $\mathbf{z}^{l+1}$ is detached to prevent updates:

$$L_{\text{pred}}^l = -\log \frac{\exp\left(\text{sim}(\hat{\mathbf{z}}^{l+1}, \mathbf{z}^{l+1})\right)}{\exp\left(\text{sim}(\hat{\mathbf{z}}^{l+1}, \mathbf{z}^{l+1})\right) + \sum_j \exp\left(\text{sim}(\hat{\mathbf{z}}^{l+1}, \mathbf{z}_{\text{neg},j}^{l+1})\right)}. \tag{1}$$

where $\mathbf{z}^{l+1}$ is the positive sample, and the set $\{\mathbf{z}_{\text{neg},j}^{l+1}\}$ contains negative samples that act as distractors. Negative samples are constructed from alternative trajectory segments generated by an LLM (GPT-4o-mini). They are plausible yet semantically misaligned with the target due to incorrect action ordering or subgoal realization. After being regularized by supervision, $\mathbf{z}^l$ serves as the input to the action-generator, where the representation of the latest time step is passed through an MLP head to produce discrete action logits.

### 3.3 PROMPT-TRAJECTORY ALIGNMENT

We introduce two complementary modules for prompt–trajectory alignment. A Sinkhorn-based alignment leverages Optimal Transport (OT) (Peyré & Cuturi, 2019) to align trajectories and prompts. This alignment operates at the distribution level, and it captures global consistency across the entire trajectory. Meanwhile, the Score-field alignment learns corrective gradients to provide step-wise guidance and adjust fine-grained executions toward higher-level semantic intent.

#### 3.3.1 SINKHORN-BASED ALIGNMENT

The Sinkhorn-based alignment module uses a distributional alignment approach, enabling the entire trajectory to align with the task's semantic structure without requiring exact token-by-token matching. The resulting gradients are dominated by aggregate semantic consistency, preventing locally ambiguous or misordered executions from influencing the alignment signal. To quantify the distributional discrepancy between the trajectory and the prompt, we employ the entropy-regularized optimal transport distance. It takes the trajectory distribution $\mu$ and the prompt distribution $\nu$ as input, and outputs a distributional alignment loss $L_{\text{sinkhorn}}(\mu, \nu)$. Formally, the Sinkhorn divergence is defined as:

$$L_{\text{sinkhorn}}(\mu, \nu) = OT_\epsilon(\mu, \nu) - \tfrac{1}{2}OT_\epsilon(\mu, \mu) - \tfrac{1}{2}OT_\epsilon(\nu, \nu), \qquad (2)$$

where $OT_\epsilon$ denotes the entropy-regularized OT cost between distributions. Minimizing $L_{\text{sinkhorn}}$ encourages the trajectory distribution $\mu$ to align semantically with the prompt distribution $\nu$ in the latent space.

#### 3.3.2 SCORE-FIELD ALIGNMENT

The Score-field module provides a local objective that complements the global Sinkhorn alignment by supplying fine-grained corrective gradients. Both $\nu$, used for Sinkhorn alignment, and the prompt embedding $\mathbf{p}$ are derived from the same prompt encoder. The Score-field module takes state embeddings $\mathbf{z}^l$ and $\mathbf{p}$ as input, and outputs localized corrective gradients $s_\psi(\mathbf{z}^l, \mathbf{p})$. The score network is implemented as an MLP. The network perturbs $\mathbf{z}^l$ with Gaussian noise $\boldsymbol{\xi} \sim \mathcal{N}(0, \sigma^2\mathbf{I})$ and learns to predict the denoising score $-\boldsymbol{\xi}/\sigma^2$.

$$L_{\text{score}} = \mathbb{E}_{(\mathbf{z}^l, \mathbf{p}), \boldsymbol{\xi} \sim \mathcal{N}(0, \sigma^2\mathbf{I})} \left[ \left\| s_\psi(\mathbf{z}^l + \boldsymbol{\xi}, \mathbf{p}) - \left( -\boldsymbol{\xi}/\sigma^2 \right) \right\|_2^2 \right]. \qquad (3)$$

It trains $s_\psi$ to model a vector field that points towards high-density regions of the prompt-defined distribution. Consequently, any trajectory state $\mathbf{z}^l$ that lies in a low-density region indicating a deviation from the prompt's semantic intent receives a corrective gradient from $s_\psi(\mathbf{z}^l, \mathbf{p})$. This gradient is incorporated as an auxiliary regularization term in the overall training objective, encouraging deviant state representations to shift toward configurations more consistent with the prompt.

### 3.4 TRAINING AND INFERENCE

In the training phase, we first pre-train the Transformer encoders with a contrastive objective. Specifically, sliding windows of state-action trajectory segments are encoded into fixed-dimensional embeddings and optimized using InfoNCE (Eq. 1). The positives are sampled from the same episode-level trajectory and negatives are generated by LLM. This stage initializes a structured embedding space that captures the plausibility and sequential dependencies of local trajectory segments.

Then we jointly fine-tune the hierarchical encoders and predictors using the following objective:

$$L_{\text{total}} = \sum_{l \in \{\text{action, subgoal}\}} \left( \lambda_{\text{pred}}^l L_{\text{pred}}^l + \lambda_{\text{score}}^l L_{\text{score}}^l \right) + \lambda_{\text{sinkhorn}} L_{\text{sinkhorn}}, \qquad (4)$$

where $\lambda_{\text{pred}}^l = 0.5$ and $\lambda_{\text{score}}^l = 0.2$ for action and subgoal levels, and $\lambda_{\text{sinkhorn}} = 0.1$. We optimize ReCAPA by jointly training the hierarchical encoders, predictors, and alignment modules under $L_{\text{total}}$. All modules are updated end-to-end, where predictive supervision from higher-level representations and prompt–trajectory alignment jointly shape lower-level representations. $\mathbf{z}^{l+1}$ used as the prediction target is detached, so gradients do not propagate to the higher-level encoder.

At inference, an LLM (GPT-4o-mini) provides a sequence of subgoals and their completion criteria, which define the candidate subtasks and termination signals. At the action level, we select Top-$K$ candidate actions and compute their alignment with the current subgoal. A candidate is accepted only if its subgoal-alignment similarity exceeds a threshold; if no candidate is accepted, the threshold is progressively relaxed and the process is retried. If all retries fail, we fall back to selecting the highest-logit action.

At the subgoal level, a sliding window encodes recent state–action sequences into a window representation. The semantic similarity between the current window representation and the embeddings of the current and next subgoals is computed. A switch to the next subgoal is triggered when alignment with the current subgoal falls below a threshold and alignment with the next subgoal exceeds it by a margin.

At the trajectory level, each candidate action is re-evaluated by temporarily appending it to the current trajectory buffer and computing prompt–trajectory consistency using a Sinkhorn-based alignment. The candidate with the highest trajectory score is selected, rather than the first candidate that passes a fixed threshold. If the highest trajectory score remains below a predefined threshold, the method falls back to the action-level top choice based on combined logits and similarity.

### 3.5 ERROR PROPAGATION METRICS

Standard metrics such as SR or Success weighted by Path Length measure whether a task is eventually completed, but they do not fully capture how errors accumulate or dissipate during execution. In long-horizon reasoning, this distinction is critical: two agents may achieve the same final success rate, yet one suffers from cascading failures while the other recovers from early slips. Without tracking such dynamics, existing metrics can mask important differences in robustness. To address this gap, we introduce two formal measures that characterize error propagation and recovery.

**Error Propagation Rate (EPR).** Let $e_t \in \{0, 1\}$ denote a step-level error indicator. $e_t = 1$ indicates that the agent's action at step $t$ violates the task constraints or deviates from the oracle trajectory, and $e_t = 0$ indicates a correct step. The EPR at lag $k$ is defined as:

$$\text{EPR}_k = \Pr(e_{t_0+k} = 1 \mid e_{t_0} = 1) - \Pr(e_{t_0+k} = 1 \mid e_{t_0} = 0), \tag{5}$$

where $\Pr(e_{t_0+k} = 1 \mid \cdot)$ denotes the probability that an error occurs at step $t_0 + k$. For example, $\text{EPR}_3 = 0.4$ means the probability of another error three steps later increases by 40% compared to the case without an initial error. Moreover, under sufficient trajectory coverage and proper conditioning, the estimator $\widehat{\text{EPR}}_k$ is consistent, i.e., $\widehat{\text{EPR}}_k \xrightarrow{p} \text{EPR}_k$.

**Propagation Attenuation Coefficient (PAC).** PAC is directly defined as:

$$\text{PAC} = -\text{slope}(\Delta, \ln \Pr\left(e_{t_0+\Delta} = 1 \mid e_{t_0} = 1\right)), \tag{6}$$

where $\text{slope}(\Delta, \cdot)$ denotes the slope obtained by fitting a least-squares linear regression of $\ln \Pr(e_{t_0+\Delta} = 1 \mid e_{t_0} = 1)$ on the lag $\Delta$ over a fixed range of $\Delta$ values. It measures the exponential decay rate of post-error risk: larger values indicate quicker recovery, while smaller values reveal that the system remains exposed to error accumulation.

## 4 EXPERIMENTS AND RESULTS

### 4.1 EXPERIMENTAL SETUP

To address trajectory drift and long-horizon planning, we evaluate ReCAPA on three benchmarks. AI2-THOR provides 120 interactive scenes while MineDojo is a Minecraft-based benchmark with 3,142 tasks. VisualAgentBench includes OmniGibson (household tasks) and Minecraft (navigation/crafting), measured by Average Success Rate (AVG) and F1. In addition to standard metrics, we report EPR and PAC to quantify error spread and recovery.

In ablations, the baseline w/o HPCC removes the HPCC module, while HPCC-AS (Action+Subgoal), HPCC-AT (Action+Trajectory), and HPCC-ST (Subgoal+Trajectory) use only two-level combinations; HPCC-Full includes all three. PPO replaces HPCC with Proximal Policy Optimization (Schulman et al., 2017), serving as a flat RL baseline. We also implement HIRO with

Table 1: Performance on AI2-THOR(Nayak et al.) across models and metrics. AI2-THOR assessed via Success Rate (SR), Transport Rate (TR), Coverage, and Balance; Coverage measures successful interactions, while Balance captures the evenness of contributions to subtasks.

| Model | SR | TR | Coverage | Balance |
|---|---|---|---|---|
| **Single-LM/Agent Baselines** | | | | |
| ReAct (Yao et al., 2023b) | 0.34 | 0.72 | 0.92 | 0.67 |
| CoT (Wei et al., 2023) | 0.14 | 0.59 | 0.87 | 0.62 |
| SmartLLM (Kannan et al., 2024) | 0.11 | 0.23 | 0.91 | 0.45 |
| CoELA (Zhang et al., 2023) | 0.25 | 0.46 | 0.76 | 0.73 |
| **Multi-Modal/LLM-Enhanced Baselines** | | | | |
| GPT-4o (Hurst et al., 2024) | 0.51 | 0.85 | 0.95 | 0.83 |
| LLaVA (Liu et al., 2023) | 0.54 | 0.84 | 0.91 | 0.75 |
| IDEFICS-2 (Laurençon et al., 2024) | 0.57 | 0.86 | 0.94 | 0.78 |
| CogVLM (Wang et al., 2024) | 0.61 | 0.89 | 0.95 | 0.80 |
| GPT-4V (Achiam et al., 2023) | 0.66 | 0.91 | **0.97** | 0.82 |
| LLaMAR (Nayak et al., 2024) | 0.68 | 0.90 | 0.95 | 0.85 |
| **ReCAPA** | **0.75** | **0.93** | 0.95 | **0.93** |

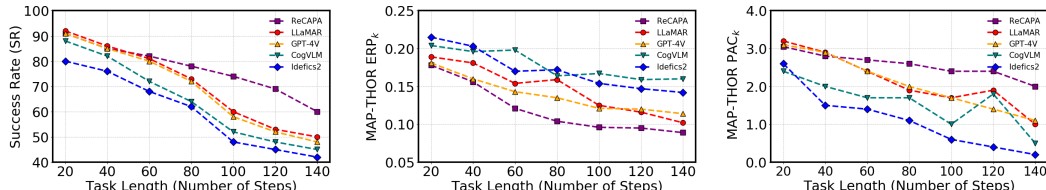

Figure 3: Left: Success rate curves across varying task lengths. Middle: EPR trends showing error propagation at different lags. Right: PAC decay rates on AI2-THOR.

two-level subgoal control, augmented with Sinkhorn and Score-field alignment. The baseline w/o Alignment removes all alignment losses, while Alignment-Full includes both. KL+Score-field replaces Sinkhorn with KL divergence while retaining Score-field.

On VisualAgentBench and AI2-THOR, we emphasize cross-domain transfer, pre-training on Proc-THOR (Deitke et al., 2022) and Behavior1K (Li et al., 2024a) and directly evaluating without fine-tuning. On MineDojo, the model is trained on all programmatic tasks except those used for testing. All baselines follow their original protocols. Further details are in Appendix A.

## 4.2 RESULTS AND ANALYSIS

Table 1, 2 and Table 3 summarize results across benchmarks. ReCAPA shows strong multi-task performance with an AVG. score of 58.65, excelling in manipulation and crafting on VisualAgent-Bench. On AI2-THOR, ReCAPA surpasses baselines with the highest SR of 0.75, TR of 0.93, and Balance of 0.93, though Coverage lags slightly behind GPT-4V. On MineDojo, ReCAPA outperforms prior LLM agents, leading in 8 out of 10 long-horizon tasks with higher success rates. It also achieves the lowest $EPR_k$ and most favorable PAC trajectory as task length grows. For example, at $k = 10$ on OmniGibson, $EPR_{10}$ is 0.082. In contrast, GPT-4o-mini and Gemini-2.5 exhibit values around 0.3, while Claude-4-sonnet exceeds 0.453. Residual errors dissipate fastest for ReCAPA, reflected in the strongest PAC value in Figure 4. Additional comparisons appear in Appendix B.

To further assess the contributions of ReCAPA's core components, we conduct ablation studies targeting the HPCC and prompt-trajectory alignment modules across four different benchmarks as shown in Table 4. Removing HPCC leads to the largest performance drop, with SR on Behavior falling to 59.3 compared to 72.2 for HPCC-Full, confirming the importance of multi-level predictive. HIRO reaches 63.4 on Behavior and 62.7 on VirtualHome, higher than PPO at 60.2 and 60.6, but generally underperforms HPCC variants. Trajectory-level variants show clear gains, with HPCC-

Table 2: Performance of different models on VisualAgentBench which include OmniGibson and Minecraft. AVG. denotes the overall average score.

| Model | AVG. | OmniGibson | Minecraft |
|---|---|---|---|
| **Open-LMMs (Fine-tuning)** | | | |
| Qwen-VL (Bai et al., 2023) | 9.90 | 1.7 | 18.1 |
| CogVLM2 (Hong et al., 2024) | 13.55 | 6.6 | 20.5 |
| LLaVA-NeXT (Li et al., 2024b) | 16.60 | 9.4 | 23.8 |
| GLM-4V (GLM et al., 2024) | 14.35 | 8.8 | 19.9 |
| InternVL-2 (Chen et al., 2024) | 22.20 | 16.0 | 28.4 |
| **Proprietary-LMMs (Prompting)** | | | |
| qwen-vl-max (Bai et al., 2023) | 2.65 | 0.0 | 5.3 |
| Claude-3.5-Sonnet (Anthropic, 2024a) | 40.15 | 24.3 | 56.0 |
| GPT-4V (preview) (Achiam et al., 2023) | 41.95 | 36.5 | 47.4 |
| GPT-4o (Hurst et al., 2024) | 48.30 | 41.4 | 55.2 |
| Claude-4-Sonnet (Anthropic, 2025) | 50.25 | 42.6 | 57.9 |
| GPT-4o mini (Zhu et al., 2023) | 54.15 | 46.7 | 61.6 |
| Gemini 2.5 Flash (Comanici et al., 2025) | 53.00 | 43.9 | 62.1 |
| **ReCAPA (Our work)** | **58.65** | **50.6** | **66.7** |

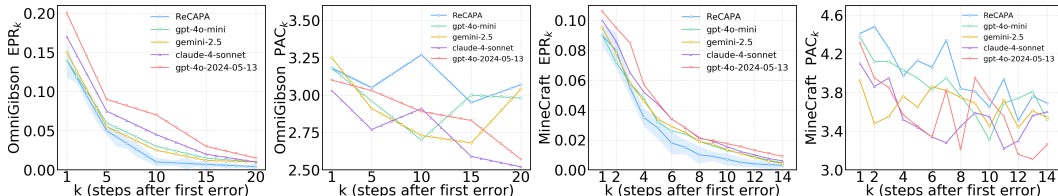

Figure 4: Results on VisualAgentBench. The left two plots show the EPR and PAC curves on OmniGibson, while the right two plots show the EPR and PAC curves on MineCraft. Shaded regions indicate 95% confidence intervals across three random seeds.

AT achieving 0.73 on AI2-THOR and HPCC-ST 0.69, both stronger than HPCC-AS. For alignment, Sinkhorn and Score-field are complementary: Sinkhorn alone gives higher scores than Score-field most of the time, and using both together achieves the best overall performance. KL+Score-field achieves the highest score of 67.0 on the Minecraft task in VisualAgentBench.

## 4.3 DISCUSSION

Our results show that ReCAPA achieves higher success rates and generally outperforms strong proprietary and open-source LMMs across benchmarks. On VisualAgentBench and MineDojo, its advantage is clearest in compositional reasoning tasks, where it decomposes goals into valid subgoals and maintains multi-step consistency. At its core, this structure continuously filters out actions that are locally plausible yet harmful in the long run, preventing small deviations from accumulating into irreversible drift. On AI2-THOR, it demonstrates stronger robustness in long-horizon planning and balanced manipulation across diverse scenes. The lower coverage relative to GPT-4V arises because ReCAPA's hierarchical favors structural consistency and high-confidence interactions, while GPT-4V's broader exploration touches more objects. Hierarchical structure improves long-horizon stability, but it can reduce coverage by being more conservative in exploration. By contrast, exploration-driven methods increase coverage and discovery by broadly probing the environment, but their reliance on trial-and-error can result in reduced completion under fixed step budgets. This reflects a fundamental trade-off in embodied agents: broader exploration increases coverage, while consistent enhances stability, and long-horizon reasoning requires balancing both.

Beyond overall success rates, we further analyze error propagation dynamics using our proposed EPR and PAC metrics. ReCAPA achieves the lowest EPR across benchmarks, showing that the

Table 3: Comparison of several tasks selected from the MineDojo(Fan et al., 2022) benchmark, covering simple resource gathering and multi-step synthesis or animal interactions. All reported values correspond to SR. The visual encoder is replaced with MINECLIP(Fan et al., 2022).

| TASK | | | | | | | | | |
|------|------|------|------|------|------|------|------|------|------|
| MINEAGENT (YU ET AL., 2024) | 0.00 | 0.00 | 0.00 | 0.00 | 0.00 | - | - | - | - |
| MINEAGENT (AUTOCRAFT) | 0.00 | 0.03 | 0.00 | 0.00 | 0.46 | 0.50 | 0.33 | 0.35 | 0.00 |
| PLAN4MC (YUAN ET AL., 2023) | 0.30 | 0.30 | 0.53 | 0.37 | 0.83 | 0.53 | 0.43 | 0.33 | 0.17 |
| RL-GPT (LIU ET AL., 2024) | **0.65** | **0.65** | 0.67 | 0.67 | 0.85 | 0.56 | 0.46 | 0.38 | 0.32 |
| RECAPA | 0.63 | **0.65** | **0.80** | **0.73** | **0.95** | **0.73** | **0.60** | **0.53** | **0.40** |

Table 4: This ablation study aims to address the role of layers and alignment strategies on four benchmarks (Li et al., 2024c) (Shridhar et al., 2021). All reported values correspond to SR.

| Method | EmbodiedAgentInterface | | AlfWorld | VisualAgentBench | | AI2-THOR |
|--------|----------|-------------|----------|------------|-----------|----------|
| | Behavior | VirtualHome | | OmniGibson | Minecraft | |
| w/o-HPCC | 59.3 | 60.1 | 80 | 42.7 | 56.3 | 0.63 |
| PPO | 60.2 | 60.6 | 79 | 41.5 | 57.8 | 0.59 |
| HIRO | 63.4 | 62.7 | 94 | 44.0 | 60.2 | 0.63 |
| HPCC-AS | 63.6 | 61.4 | 86 | 43.4 | 62.5 | 0.65 |
| HPCC-AT | 65.1 | **70.9** | 94 | 47.9 | 57.5 | 0.73 |
| HPCC-ST | 66.3 | 66.3 | 91 | 48.1 | 60.4 | 0.69 |
| HPCC-Full | **72.2** | 70.5 | **96** | **50.6** | **66.7** | **0.75** |
| w/o-Alignment | 65.8 | 67.2 | 92 | 46.1 | 62.4 | 0.69 |
| Sinkhorn | 66.1 | 69.4 | 95 | 49.3 | 65.6 | 0.74 |
| Score-field | 64.4 | 67.9 | 92 | 46.8 | 66.3 | 0.72 |
| KL + score-field | 70.3 | 68.1 | 95 | 49.6 | **67.0** | 0.74 |
| Alignment-Full | **72.2** | **70.5** | **96** | **50.6** | 66.7 | **0.75** |

impact of early mistakes dissipates more quickly than in other LMMs. This indicates that while errors remain, ReCAPA limits their spread and prevents small deviations from escalating into full failures. Similarly, ReCAPA maintains the highest PAC trajectory, indicating that errors dissipate more rapidly, and longer tasks provide more opportunities for recovery rather than compounding drift. Taken together, the two metrics highlight complementary aspects of robustness, with low EPR reflecting error prevention and high PAC reflecting error recovery, which prior evaluations often overlooked. Standard success-based metrics conflate these effects by only measuring terminal outcomes, masking whether failures arise from early error amplification or poor recovery. More broadly, EPR and PAC provide a useful lens for analyzing long-horizon reasoning and may encourage future evaluation to move beyond stepwise accuracy toward explicitly quantifying how agents prevent and dissipate cascading errors.

In ablation studies, by linking global trajectories with local actions or subgoals, HPCC-AT and HPCC-ST reduce the drift of locally steps from global goals. Trajectory-level representations act as a global semantic reference that selectively suppresses local updates misaligned with the overall goal Zeng et al. (2026). This suggests that effective long-horizon reasoning requires cross-level guidance, whereas LLMs, though large, often optimize only for local coherence and struggle to maintain consistency over extended horizons. HIRO executes fixed-interval subgoals open-loop, so when the environment shifts it lacks flexible adjustments and actions drift from global goals easily. HPCC instead introduces an adaptive correction strategy, where cross-level feedback adjust local actions and keep them aligned with global goals. For alignment, KL+Score-field's sensitivity strongly penalizes minor mismatches. In Minecraft, where distributions are skewed, this sensitivity helps capture rare but decisive events. However, it also destabilizes signals, making KL-based alignment less reliable than Alignment-Full, which achieves stronger overall performance across most tasks. This highlights a trade-off between sensitivity and stability in alignment objectives: overly sensitive divergences can capture rare signals but risk amplifying noise in long-horizon settings.

## 5 CONCLUSION

To mitigate semantic drift in long-horizon reasoning for embodied agents, we proposed ReCAPA, a framework that integrates hierarchical correction with prompt-trajectory alignment. Experiments on VisualAgentBench, MineDojo and AI2-THOR both demonstrate that ReCAPA outperforms strong baselines. While ReCAPA achieves SOTA results across AI2-THOR, VisualAgentBench and Mine-Dojo, it exhibits two key limitations. (1) The correction mechanism operates through discrete scoring at the hierarchical levels, which stabilizes execution trajectory but cannot provide continuous, intermediate feedback. As a result, small deviations that accumulate between scoring steps may not be corrected immediately. (2) The hierarchical generation module uses deterministic mappings to compute next-layer embeddings, meaning the model follows a single subgoal trajectory and cannot represent alternative plausible continuations when uncertainty is high. To address these limitations, future work will explore mechanisms that provide richer intermediate feedback and better capture uncertainty in hierarchical reasoning. Our goal is to enable more flexible long-horizon decision making without being restricted to a single deterministic trajectory.

## 6 ACKNOWLEDGMENTS

This work is supported by the National Natural Science Foundation of China (No. 62406267), Guangdong Provincial Project (No. 2024QN11X072) and Guangzhou Municipal Science and Technology Project (No. 2025A04J4070).

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

## A    DETAILED EXPERIMENTAL PROCEDURES

ReCAPA employs a two-stage training process. The first stage is offline pre-training on diverse expert and LLM-generated trajectories to establish hierarchical predictive and alignment capabilities. In the second stage, we adopt benchmark-specific protocols: some benchmarks (e.g., MineDojo) involve supervised adaptation using in-domain trajectories, while others (e.g., VisualAgentBench and AI2-THOR) emphasize pure cross-domain transfer without task-specific fine-tuning.

### A.1    STAGE 1: OFFLINE PRE-TRAINING

The offline pre-training phase leverages expert demonstration trajectories to establish a foundational understanding of embodied interaction and planning for ReCAPA. This phase is critical for equipping the model with the necessary knowledge to generalize across tasks before any domain-specific fine-tuning is applied. We utilize two distinct datasets, `BEHAVIOR-1K` and `ProcTHOR`, to cover a broad range of scenarios and ensure diverse task representation.

- **BEHAVIOR-1K:** This benchmark focuses on complex, everyday human activities in simulated environments. For this dataset, we select 300 representative tasks and use 3 to 5 expert demonstration trajectories per task. The fine-grained interaction data allows the model to learn the dynamics of typical human behavior in an embodied context.

- **ProcTHOR:** In addition to `BEHAVIOR-1K`, we incorporate 300 diverse tasks from the procedurally generated `ProcTHOR` environment. This dataset features various scene layouts and object arrangements, and we collect approximately 10 expert trajectories for each task. The diversity of this dataset ensures the model is not overly dependent on specific environmental configurations, enhancing its generalization ability.

This foundational pre-training step is conducted in an offline manner, allowing the model to absorb critical information about trajectories without the complexity of real-time interaction. Although the total number of trajectories is smaller than in large-scale pretraining corpora, the diversity across BEHAVIOR-1K and ProcTHOR ensures broad task coverage and equips ReCAPA with transferable knowledge of embodied dynamics. This design emphasizes sample efficiency and cross-domain generalization: in Stage 2, the model is evaluated both with limited in-domain adaptation (e.g., MineDojo) and under pure transfer settings (e.g., VisualAgentBench and AI2-THOR), highlighting ReCAPA's ability to generalize beyond its training distribution.

### A.2    STAGE 2: DOMAIN-SPECIFIC ADAPTATION AND TRANSFER

Following the pre-training phase, we distinguish two types of evaluation protocols. On MineDojo, the model is trained on all programmatic tasks except those used for testing.

For VisualAgentBench, AI2-THOR and EmbodiedAgentInterface, we emphasize pure cross-domain transfer: the model is pre-trained on ProcTHOR and BEHAVIOR-1K and directly evaluated on the target benchmarks without task-specific fine-tuning. This setting highlights ReCAPA's generalization ability under strict transfer conditions. Baselines are evaluated under their original protocols to ensure fair comparison.

### A.3    TRAINING CONFIGURATION

For all environments, we use the AdamW optimizer, with a learning rate of 1e-4, a weight decay of 0.01, and a batch size of 32. The learning rate follows a cosine schedule with a linear warm-up over the first 1,000 steps. Training is conducted for a total of 200,000 steps, with the model being trained on 4 NVIDIA H20 GPUs to ensure efficient scaling across large datasets. The text encoder (nomic-embed-text-v1.5) encodes prompt tokens, while the vision encoder (nomic-embed-vision-v1.5) encodes environmental observations.

In summary, the two-stage training protocol, comprising offline pre-training and online fine-tuning, equips ReCAPA with the capacity to handle complex, long-horizon tasks in varied environments. The incorporation of expert trajectories during pre-training and the dynamic, memory-augmented

Table 5: PAC vs. Task Length: Measures the attenuation rate of the impact of early errors on subsequent steps (the higher the value, the faster the recovery and the weaker the error propagation)

| Model | 20 | 40 | 60 | 80 | 100 | 120 | 140 |
|---|---|---|---|---|---|---|---|
| **ReCAPA** (Our work) | 3.1 | 2.8 | 2.7 | 2.6 | 2.4 | 2.2 | 2.0 |
| **LLaMAR** (Nayak et al., 2024) | 3.2 | 2.9 | 2.4 | 1.9 | 1.7 | 1.4 | 1.0 |
| **GPT-4V** (Achiam et al., 2023) | 3.1 | 2.9 | 2.4 | 2.0 | 1.7 | 1.4 | 1.1 |
| **CogVLM** (Wang et al., 2024) | 2.4 | 2.0 | 1.7 | 1.4 | 1.0 | 0.8 | 0.5 |
| **IDEFICS-2** (Laurençon et al., 2024) | 2.6 | 1.5 | 1.2 | 1.1 | 0.6 | 0.3 | 0.0 |

Table 6: Performance of ReCAPA Variants under Layer-wise Ablation across Benchmarks. This table estimates the expected impact of removing individual layers or flattening the hierarchy (Flat-Head) on ReCAPA's performance across diverse environments, highlighting the necessity of each abstraction level

| Method | EmbodiedAgentInterface | | AlfWorld | VisualAgentBench | | AI2-THOR |
|---|---|---|---|---|---|---|
| | **Behavior** | **VirtualHome** | | **OmniGibson** | **Minecraft** | |
| w/o Subgoal-Level | 62.8 | 61.5 | 87 | 44.8 | 61.4 | 0.62 |
| w/o Trajectory-Level | 65.4 | 64.5 | 86 | 47.4 | 60.5 | 0.67 |
| FlatHead | 49.3 | 55.4 | 78 | 35.6 | 44.1 | 0.52 |
| ReCAPA | **72.2** | **70.5** | **96** | **50.6** | **66.7** | **0.75** |

fine-tuning mechanism ensures that ReCAPA continuously improves its performance and generalizes effectively to unseen scenarios. As shown in Table 5, it presents the PAC results on AI2-THOR, showing how different models recover from early errors across varying task lengths.

To assess the independent contribution of each hierarchical level in ReCAPA's two-level structure (subgoal → trajectory), we conduct a layer-wise ablation study. While the full model integrates across all levels, it remains unclear whether each level provides unique benefits. In this experiment, we selectively remove one module at a time—trajectory-level or subgoal-level, keeping the rest of the architecture intact. This design allows us to evaluate whether each layer offers distinct abstraction or semantic supervision, whether model performance depends on cross-scale reasoning, and whether the joint use of all levels outperforms partial configurations. In addition, we introduce a FlatHead variant, which collapses the entire hierarchy into a single decoder without explicit levels, to test whether the hierarchical structure itself is essential for semantic reasoning.

## B  COMPLEMENTARY RESULTS AND ANALYSIS

### B.1  LAYER ABLATION

The ablation results in Table 6 highlight the necessity of maintaining a multi-level structure in ReCAPA. Removing any individual layer leads to a consistent drop in performance across all benchmarks, confirming that each level contributes uniquely to hierarchical reasoning. Notably, the removal of the mid-level results in the most severe degradation, especially on Behavior (–9.4 points) and VirtualHome (–9.0), suggesting that the mid-level layer plays a critical role in bridging trajectory-level plans with low-level execution. The w/o Trajectory-Level variant also suffers substantial loss in ALFWorld and AI2-THOR, demonstrating that long-horizon environments with sparse rewards and delayed feedback heavily depend on long-range planning. The FlatHead baseline, which removes the entire hierarchy, performs the worst across all benchmarks—underscoring the indispensable value of structured abstraction and layered semantic supervision for complex task

Table 7: Comparative Evaluation of ReCAPA and Hierarchical Variants: Effects of Coupling and Alignment Strategy. ReCAPA integrates all levels via joint and multi-scale alignment, allowing semantic corrections to propagate throughout the hierarchy

| Method | EmbodiedAgentInterface | | AlfWorld | VisualAgentBench | | AI2-THOR |
| --- | --- | --- | --- | --- | --- | --- |
| | Behavior | VirtualHome | | OmniGibson | Minecraft | |
| Separate–BottomUp | 62.1 | 62.7 | 79 | 43.9 | 57.9 | 0.59 |
| Separate–Parallel | 63.4 | 60.8 | 76 | 45.4 | 60.5 | 0.54 |
| Separate–TopDown | 66.7 | 64.5 | 86 | 47.4 | 62.5 | 0.67 |
| Frozen Traj-Level | 68.9 | 68.0 | 90 | 48.3 | 63.9 | 0.71 |
| ReCAPA | **72.2** | **70.5** | **96** | **50.6** | **66.7** | **0.75** |

generalization. These findings validate ReCAPA's core design principle: performance in long-horizon embodied tasks emerges from coordinated across abstraction levels.

## B.2 COUPLING STRATEGY COMPARISON

To further examine the role of joint optimization across hierarchical layers, we introduce several additional ablation variants targeting the training strategy of HPCC:

- **Frozen Traj-Level**: The Traj-level module remains active during forward execution but its parameters are frozen throughout training. Only the mid- and low-level modules are updated, isolating the contribution of Traj-level gradient signals.

- **Separate–BottomUp**: Each layer is trained sequentially in a bottom-up manner: the action-level module is trained first, then frozen; the subgoal-level is trained next with action-level frozen; finally, the trajectory-level module is trained on top. This mimics a stage-wise curriculum from action primitives to subgoal planning.

- **Separate–TopDown**: The reverse of BottomUp. Training proceeds from the trajectory-level module down to the action-level, with each previously trained module frozen at its respective stage. This configuration reflects top-down reasoning pipelines, starting from global goals to execution-level commands.

- **Separate–Parallel**: All three layers are trained independently without inter-layer gradient flow. Each module is optimized on its respective sub-task using its local alignment and trajectory data. This configuration serves as a strong baseline to test whether cross-layer interactions are necessary.

Each of these configurations is trained under the same data regime and alignment loss structure (Sinkhorn + Score Field), allowing for controlled comparisons with the jointly optimized ReCAPA model. We report task-level metrics to evaluate performance degradation and convergence stability.

Table 7 presents a comparative study of different hierarchical training strategies across six embodied benchmarks. ReCAPA achieves the highest performance on all tasks, demonstrating the benefit of fully joint training across three corrective modules. In contrast, all decoupled baselines underperform to varying degrees, each revealing critical weaknesses in alternative optimization schemes.

The Separate–BottomUp configuration performs the worst overall, as it trains low-level modules in isolation before exposing them to task objectives. This leads to suboptimal primitive behaviors that constrain downstream learning and confirms that without task-aware supervision results in inefficient or misaligned action policies. The Separate–TopDown baseline performs moderately better , but still lags behind ReCAPA. Despite training from task goals downward, the lack of feedback from lower layers causes the top-level planner to overfit to idealized subgoal sequences that may not align with actual execution capabilities, which results in a "planning-execution mismatch." The Separate–Parallel setting confirms this issue from another angle: although each module becomes competent in isolation, the lack of cross-layer adaptation leads to representational inconsistency and

Table 8: This ablation study evaluates the impact of prediction and components in HPCC by isolating the effect of removing prediction, , or both.

| Variant | EmbodiedAgentInterface | | AlfWorld | VisualAgentBench | | AI2-THOR |
| | Behavior | VirtualHome | | OmniGibson | Minecraft | |
| --- | --- | --- | --- | --- | --- | --- |
| w/o pred and refl | 62.6 | 63.1 | 85 | 44.5 | 60.5 | 0.66 |
| Only Prediction (no refl) | 65.4 | 66.3 | 90 | 47.8 | 60.9 | 0.72 |
| Only (no pred) | 67.1 | 64.9 | 91 | 49.2 | 62.0 | 0.72 |
| Full HPCC (pred + refl) | 72.2 | 70.5 | 96 | 50.6 | 66.7 | 0.75 |

semantic drift between layers. The resulting interface mismatch limits coordination across hierarchical stages. Frozen Traj-Level, which freezes the top-level module and only updates the mid- and low-level components, yields decent performance, but falls short of ReCAPA. This highlights that trajectory-level goal representations also require continual adaptation to downstream dynamics in order to maintain semantic coherence.

Taken together, these results empirically validate our theoretical claim that hierarchical must be jointly optimized to achieve sample-efficient and semantically aligned behavior. The observed performance gaps support our convergence analysis under multi-objective optimization and further justify the design of ReCAPA's multi-level update strategy.

### B.3 Ablation on Prediction and Alignment

To further disentangle the respective contributions of prediction and within HPCC, as shown in Table 8 we designed four ablation variants. The first, w/o pred and refl, removes both components entirely, leaving only the hierarchical execution head to generate actions and serving as a baseline without any auxiliary consistency signals. The second, Only Prediction, retains the forward prediction modules that forecast higher-level trajectory or subgoal embeddings to regularize lower-level execution, but discards the modules that would otherwise check and realign actions during rollouts; this setting isolates the benefit of anticipatory guidance without any corrective feedback. The third, Only , removes prediction while preserving the consistency checks and corrective updates after each execution step, thereby examining the effect of purely reactive recovery in the absence of foresight. Finally, Full HPCC activates both prediction and to form a closed loop in which lower-level policies both anticipate higher-level representations and immediately repair inconsistencies as they arise. Comparing these four settings allows us to characterize the distinct roles of prediction (proactive prevention) and (reactive correction), as well as their synergy when combined.

This ablation highlights the distinct yet complementary roles of prediction and within HPCC. Eliminating both modules severely degrades performance, confirming that hierarchical execution heads alone are insufficient for mitigating long-horizon drift. When only prediction is retained, the agent benefits from anticipatory alignment signals that reduce short-horizon inconsistencies (e.g., improved results on VisualAgentBench), but the absence of reactive correction allows early errors to cascade unchecked, resulting in limited gains on AlfWorld and AI2-THOR. Conversely, keeping only provides the ability to recover after errors occur, which is particularly beneficial for tasks with extended horizons where error accumulation dominates; however, without forward prediction, the system lacks proactive guidance and remains vulnerable to subtle misalignments in representation space. Full HPCC consistently outperforms all ablated variants because it unifies both mechanisms: prediction serves as an early warning system that lowers the likelihood of compounding failures, while functions as a recovery channel that actively attenuates error propagation.

### B.4 Evaluation on Embodied Agent Interface

As show in Table 9, it evaluates ReCAPA and competing models on the Embodied Agent Interface benchmark, focusing on Behavior and VirtualHome tasks. The metrics include Performance (Perf.), Goal F1, Action Sequence accuracy (both Task and Execution levels), and Subgoal Decomposition (Task and Execution), with an overall F1 score. ReCAPA achieves robust performance (72.2 Perf.,

Table 9: Results on the Behavior task of the Embodied Agent Interface benchmark. ReCAPA shows strong performance with leading scores in Goal F1 (84.8), Action Sequencing (77.0/84.0), and Subgoal Decomposition (53.0/60.0), surpassing baselines such as o1-preview and Claude-4. These results highlight its effectiveness in long-horizon reasoning and precise subgoal planning through hierarchical architecture and alignment mechanisms

| Models | Perf. | Goal F1 | Action Seq. | | Subgoal Dec. | | F1 |
|--------|-------|---------|------|------|------|------|-----|
| | | | Task | Exec. | Task | Exec. | |
| **o1-preview** Zhong et al. (2024) | **74.9** | 81.6 | **81.0** | **91.0** | **57.0** | **62.0** | **70.8** |
| ReCAPA | 72.2 | **84.8** | 77.0 | 84.0 | 53.0 | 60.0 | 69.8 |
| Claude-4-Sonnet Anthropic (2024) | 68.5 | 84.3 | 68.0 | 75.0 | 47.0 | 52.0 | 69.2 |
| Claude-3.5-Sonnet Anthropic (2024a) | 64.2 | 82.7 | 60.0 | 69.0 | 39.0 | 44.0 | 67.9 |
| Claude-3-Opus Anthropic (2024b) | 60.4 | 77.0 | 51.0 | 59.0 | 41.0 | 47.0 | 63.4 |
| GPT-4o Hurst et al. (2024) | 59.8 | 79.2 | 47.0 | 53.0 | 49.0 | 55.0 | 60.9 |
| o1-mini Jaech et al. (2024) | 57.5 | 76.4 | 56.0 | 65.0 | 31.0 | 39.0 | 56.4 |
| Claude-3-Sonnet Anthropic (2024b) | 55.1 | 69.4 | 44.0 | 57.0 | 39.0 | 43.0 | 56.2 |
| Gemini-1.5-Flash Team et al. (2024) | 52.1 | 74.8 | 40.0 | 52.0 | 34.0 | 42.0 | 53.4 |
| Mistral-Large Jiang et al. (2023) | 50.4 | 74.3 | 33.0 | 50.0 | 31.0 | 38.0 | 49.5 |

Table 10: Results on the VirtualHome task of the Embodied Agent Interface benchmark. ReCAPA shows consistently strong performance across six metrics, leading in Overall Perf. (70.5), Subgoal Decomposition (Task SR: 94.5, Exec. SR: 91.7), and Transition Modeling (F1: 64.9, Plan SR: 84.6). Compared to o1-preview and Claude models, it demonstrates superior hierarchical reasoning and robust embodied planning in complex VirtualHome environments

| Model Family | Overall Perf. | F1 | Action Sequencing | Subgoal Decomposition | | Transition Modeling | |
|--------------|---------------|-----|-------------------|------------------------|------|---------------------|------|
| | | | Task SR | Task SR | Exec. SR | F1 | Plan SR |
| ReCAPA | **70.5** | **44.2** | 72.8 | **94.5** | **91.7** | **64.9** | 84.6 |
| Claude-4-Sonnet Anthropic (2024) | 69.1 | 43.5 | **73.6** | 92.7 | 90.9 | 51.4 | 86.8 |
| o1-preview Zhong et al. (2024) | 65.8 | 42.7 | 71.1 | 93.2 | 89.4 | 48.0 | 72.4 |
| Gemini-Pro Team et al. (2024) | 65.3 | 37.9 | 73.1 | 91.1 | 87.0 | 34.1 | **91.9** |
| Claude-3-Sonnet Anthropic (2024b) | 64.9 | 33.0 | 72.8 | 92.0 | 89.1 | 48.9 | 80.5 |
| GPT-4o Hurst et al. (2024) | 60.8 | 36.5 | 61.6 | 91.1 | 87.6 | 46.7 | 68.2 |
| Claude-3-Opus Anthropic (2024b) | 59.9 | 31.4 | 66.2 | 89.9 | 86.7 | 48.8 | 61.8 |
| o1-mini Jaech et al. (2024) | 57.9 | 31.2 | 65.9 | 84.6 | 79.3 | 41.5 | 69.0 |

84.8 Goal F1), trailing only o1-preview in Perf. but excelling in Goal F1. The training methodology for these tasks follows a two-stage protocol: offline pre-training on state-action-reward trajectories initialized via GPT-4o API, followed by benchmark-specific optimization. For Behavior tasks, ReCAPA leverages ProcTHOR's procedurally generated environments (30 tasks, 10 expert trajectories per task) to enhance scene generalization, avoiding overfitting. The model is trained end-to-end with AdamW (lr=1e-4, weight decay=0.01, batch size=32) for 200K steps on 4 NVIDIA H20 GPUs, using a cosine learning rate schedule with 1,000-step warm-up. We obtain base text and vision embeddings from pre-trained Nomic encoders (nomic-embed-text-v1.5 and nomic-embed-vision-v1.5), which remain frozen; only the HPCC and alignment modules are optimized. This approach ensures strong performance in both trajectory-level goal reasoning (84.8 F1) and action sequencing (77.0 Task, 84.0 Exec.), as reflected in the results.

As show in Table 10, it provides a comprehensive evaluation across six key metrics: Overall Performance, F1, Action Sequencing (Task SR, Exec. SR), Subgoal Decomposition (F1), and Transition Modeling (Plan SR). ReCAPA outperforms competitors in Overall Perf. (70.5) and Transition Modeling (84.6 Plan SR), demonstrating its strength in long-horizon planning and trajectory stability. The training pipeline mirrors the methodology for AI2-THOR and VisualAgentBench, emphasiz-

Table 11: Models performance on AlfWorld benchmark

| Method | Success rate (%) ↑ | | | | | | |
|---|---|---|---|---|---|---|---|
| | Avg. | Pick | Clean | Heat | Cool | Examine | Picktwo |
| **Vision-language models** | | | | | | | |
| MiniGPT-4* | 16 | 4 | 0 | 19 | 17 | 67 | 6 |
| BLIP-2* | 4 | 0 | 6 | 4 | 11 | 6 | 0 |
| LLaMA-Adapter* | 13 | 17 | 10 | 27 | 22 | 0 | 0 |
| InstructBLIP* | 22 | 50 | 26 | 23 | 6 | 17 | 0 |
| EMMA* | **82** | **71** | **94** | **85** | **83** | **88** | **67** |
| **Language models** | | | | | | | |
| BUTLER* | 26 | 31 | 41 | 60 | 27 | 12 | 29 |
| DEPS | 76 | 93 | 50 | 80 | **100** | **100** | 0 |
| AutoGen* | 77 | 92 | 74 | 78 | 86 | 83 | 41 |
| ReAct | 74 | 79 | 54 | 96 | 85 | 83 | 51 |
| AdaPlanner | 91 | 100 | **100** | 89 | **100** | 97 | 47 |
| Reflexion | 86 | 92 | 94 | 70 | 81 | 90 | 88 |
| RAFA | 95 | 100 | 97 | 91 | 95 | **100** | 82 |
| WALL-E 1.0 | 95 | 100 | 97 | **100** | 86 | 85 | **100** |
| WALL-E 2.0 | **98** | 100 | **100** | 96 | **100** | 100 | 94 |
| **ReCAPA (ours)** | **96** | **100** | **97** | 94 | 95 | 96 | 94 |

ing cross-domain transfer through few-shot pre-training on ProcTHOR (no BEHAVIOR-1K in this phase) and task-specific fine-tuning. For AlfWorld tasks, ReCAPA employs RAFA-style multi-round interactions with memory-augmented prompts to improve online adaptation. The technical setup matches Table 6's (AdamW, 200K steps), with alignment losses (Full Alignment) and hierarchical (HPCC-Full) critical to its success in Subgoal Decomposition (64.9 F1) and Action Sequencing (94.5 Exec. SR). Claude-4-Sonnet and Gemini-Pro show competitive Plan SR (86.8, 91.9), but ReCAPA balances all metrics, underscoring its versatility.

### B.5 PERFORMANCE ANALYSIS AND DISCUSSION

From the results in Table 11, our model ReCAPA consistently demonstrates clear advantages over both proprietary and open-source LMM baselines. In particular, ReCAPA shows strong performance on long-horizon tasks such as PICKTWO, where success rates remain significantly higher than all other baselines. This confirms that our hierarchical planning and mechanisms can effectively manage compositional goals that require multiple coordinated steps.

Despite its overall superiority, ReCAPA is not without limitations. WALL-E 2.0, augmented with reinforcement learning fine-tuning, achieves slightly higher scores on some subtasks. This gap reflects a trade-off: ReCAPA is primarily designed for sparse or reward-agnostic settings, which limits its ability to exploit dense feedback signals as efficiently as reinforcement learning. Overall, the results demonstrate that ReCAPA offers a balanced and generalizable solution: it consistently outperforms competitors in challenging long-horizon tasks, while only marginally trailing in AlfWorld where dense rewards provide a strong supervision signal.

## C FAILURE CASE ANALYSIS

To complement the quantitative results (EPR, PAC, and ablations), we provide qualitative analyses of typical failure modes. As shown in Table 12, these examples illustrate how cascading errors arise and how ReCAPA mitigates them compared to ablated variants.

We categorize common failure types observed during evaluation:

- **Prompt:** "Take the milk from the fridge."

| Failure Type | Description / Example |
|---|---|
| Subgoal Ordering Error | Executing subgoals in the wrong order (e.g., attempting to close the fridge before retrieving the milk). This leads to invalid or incomplete task outcomes. |
| Entity Grounding Error | Misidentifying or manipulating the wrong object (e.g., taking juice instead of the intended milk). |
| Premature Termination | Ending the task early before completing all required subgoals (e.g., retrieving the milk but leaving the fridge door open). |
| Looping / Redundancy | Performing unnecessary actions without contributing to the goal (e.g., taking an extra apple after already retrieving the milk). |

Table 12: Taxonomy of common failure types in hierarchical agent execution, adapted to the fridge–milk task with descriptions and examples.

- **Full ReCAPA:** Executes correctly by [Open fridge] → [Take milk] → [Close fridge].
- **w/o HPCC:** Retrieves the milk but then performs an unnecessary action [Take apple] and forgets to [Close fridge], leading to redundancy and premature termination.

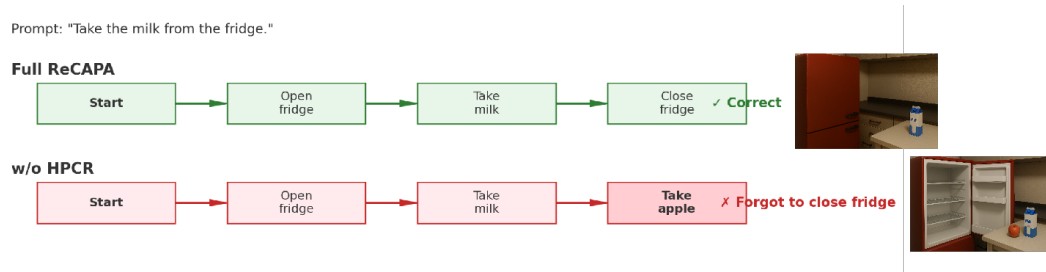

Figure 5: Representative failure case for the prompt "Take the milk from the fridge." Full ReCAPA executes correctly by opening the fridge, retrieving the milk, and closing the fridge. In contrast, the ablated model (w/o HPCC) retrieves the milk but then takes an unrelated item and forgets to close the fridge, leaving the environment in an invalid state and illustrating how local missteps cascade into task failure.

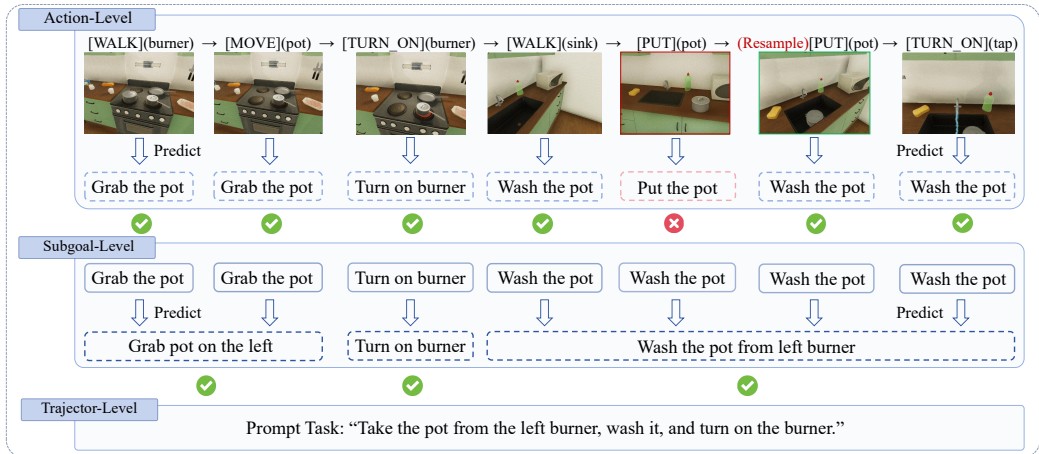

Figure 6: Illustration of ReCAPA's hierarchical correction during inference.

As shown in Figure 5, execution begins correctly as the agent approaches the pot and moves toward the sink. The action-level representation fails to predict the correct subgoal-level semantic target ("place the pot into the sink"), causing action–subgoal alignment to drop. Subgoal-level forecast-

ing then drifts toward an incorrect direction, which subsequently causes global prompt–trajectory alignment to decrease.

ReCAPA detects this cross-level inconsistency and resamples the action until the pot is correctly placed in the sink. Once the corrected action is taken, the restored alignment propagates upward: subgoal-level semantics realign with the intended target, and the trajectory-level embedding recovers. Execution then resumes and completes normally.

# D  ERROR PROPAGATION METRICS (EPR AND PAC)

## D.1  ERROR PROPAGATION RATE (EPR)

### D.1.1  DEFINITION

We define the *Error Propagation Rate (EPR)* at lag $k$ as the marginal increase in error probability $k$ steps after the first error in an episode:

$$\text{EPR}_k \;=\; \Pr(e_{t_0+k} = 1 \mid e_{t_0} = 1) \;-\; \Pr(e_{t_0+k} = 1 \mid e_{t_0} = 0),$$

where $e_t \in \{0, 1\}$ is the step-level error indicator and $t_0$ denotes the first-error time in the trajectory. Intuitively, $\text{EPR}_k$ isolates the excess risk attributable to an initial failure, relative to a matched no-error baseline.

### D.1.2  INTERPRETATION AND PROPERTIES

**Range and edge cases.**  $\text{EPR}_k \in [-1, 1]$. $\text{EPR}_k \approx 0$ implies effective containment; large positive values indicate cascading failures; negative values may indicate *active recovery*, i.e., the model becomes less error-prone after an initial mistake.

**Connection to hazard/recovery dynamics.**  If errors can be modeled by a Markovian error-state abstraction, $\text{EPR}_k$ equals the difference in $k$-step transition probabilities into the error state under two initial conditions ($e_{t_0}{=}1$ vs. 0). It is therefore a direct proxy for cascade tendency. Compared to the Propagation Attenuation Coefficient (PAC), which measures the *decay rate* of post-error risk, EPR reflects the *marginal elevation* of risk due to an initial error.

**Consistency.**  Suppose the rollout process is ergodic and the matching function $\phi(\mathcal{F}_{t_0})$ successfully blocks confounding (i.e., controls for task stage and context). Then $\widehat{\text{EPR}}_k$ converges in probability to $\text{EPR}_k$ as the number of episodes grows. If censoring arises (e.g., episodes ending before $t_0 + k$), an inverse-probability-of-censoring weighted (IPCW) estimator ensures consistency.

**Granularity.**  EPR can be computed at different levels:

- *Action-level* ($e_t^{(\text{act})}$): low-level execution slips.
- *Subgoal-level* ($e_t^{(\text{sub})}$): structural or DAG violations.

Comparing $\text{EPR}_k^{(\text{act})}$ vs. $\text{EPR}_k^{(\text{sub})}$ helps identify whether cascades are driven by local control errors or higher-level planning flaws.

### D.1.3  ESTIMATION PROTOCOL

The empirical procedure is as follows:

1. **Identify first-error times.** For each trajectory $i$, locate $t_0^{(i)}$ (the earliest $t$ such that $e_t = 1$). If no error occurs, the trajectory is excluded from case construction.

2. **Construct case and control sets.** For each $(i, t_0^{(i)})$, treat the pair $(i, t_0^{(i)})$ as a "case." Find a "control" $(j, \tilde{t})$ from another trajectory $j$ such that:

- $e_{\tilde{t}}^{(j)} = 0$ with no prior errors,

- $\tilde{t} + k \le T^{(j)}$,

- Contexts are matched: $\phi(\mathcal{F}_{\tilde{t}}^{(j)}) \approx \phi(\mathcal{F}_{t_0^{(i)}}^{(i)})$, where $\phi$ encodes subgoal ID, horizon length, scene category, and latent trajectory state.

3. **Compute probabilities.** Estimate $\widehat{p}_{\text{case}}(k) = \Pr(e_{t_0+k} = 1 \mid e_{t_0} = 1)$ and $\widehat{p}_{\text{ctrl}}(k) = \Pr(e_{\tilde{t}+k} = 1 \mid e_{\tilde{t}} = 0)$ from matched pairs.

4. **Form EPR estimate.**

$$\widehat{\text{EPR}}_k = \widehat{p}_{\text{case}}(k) - \widehat{p}_{\text{ctrl}}(k).$$

5. **Summarize across $k$.** In addition to plotting $\widehat{\text{EPR}}_k$ as a function of $k$, we report:

- AUC-EPR$_W = \sum_{k=1}^{W} \widehat{\text{EPR}}_k$ for horizons $W \in \{3, 5\}$,

- the slope of $\widehat{\text{EPR}}_k$ over $k$ as a compact one-number summary of cascade growth.

### D.1.4 REPORTING RECOMMENDATIONS

**Visualization.** Always report $\widehat{\text{EPR}}_k$ vs. $k$ with 95% confidence intervals (bootstrapped by episode).

**Summary statistics.** Report AUC-EPR$_W$ at $W = 3, 5$ as a concise summary statistic. For long-horizon tasks, include slope-of-$k$ analysis to quantify cascade growth.

**Comparisons.** When comparing models, include both absolute differences and relative reductions in EPR. Use identical $e_t$ definitions and matching hyperparameters across models to ensure fairness.

—

### D.2 PRACTICAL CONSIDERATIONS

**Censoring.** Restrict $W$ to be below the 25th percentile of remaining horizon $(T - t_0)$ to avoid heavy censoring bias.

**Variance estimation.** Use per-episode bootstrap resampling for CI bands.

**Level separation.** Always report both action-level and subgoal-level EPR curves to clarify the source of propagation.

**Robustness checks.** Verify results are stable to the choice of $\phi$ (matching function) and distance metric.

### D.3 PROPAGATION ATTENUATION COEFFICIENT (PAC)

**Definition.** We define the *Propagation Attenuation Coefficient (PAC)* at lag $k$ as the relative decay rate of post-error risk:

$$\text{PAC}_k = \frac{\Pr(e_{t_0+k} = 1 \mid e_{t_0} = 1)}{\Pr(e_{t_0+1} = 1 \mid e_{t_0} = 1)},$$

where $e_t$ is the error indicator and $t_0$ denotes the first-error time. Intuitively, PAC measures how quickly the elevated error probability induced by the first error attenuates over time. A value close to 1 indicates persistent risk, while a value below 1 indicates attenuation.

### D.3.1 INTERPRETATION AND PROPERTIES

**Range and meaning.** $\text{PAC}_k \in [0, \infty)$.

- $\text{PAC}_k \approx 1 \Rightarrow$ error risk is persistent, cascades continue.
- $\text{PAC}_k < 1 \Rightarrow$ error risk attenuates; the system recovers.
- $\text{PAC}_k > 1 \Rightarrow$ error risk escalates faster than the initial shock (rare but possible in unstable systems).

**Connection to survival/hazard analysis.** PAC can be viewed as an analogue of a hazard decay factor: it compares the conditional error hazard at lag $k$ to that immediately after the error. Whereas EPR captures the *absolute marginal risk increase*, PAC quantifies the *relative decay speed* of this risk.

**Granularity.** PAC can be applied at both:

- *Action-level*: robustness of local control after a slip.
- *Subgoal-level*: structural recovery after violating a planning dependency.

### D.3.2 ESTIMATION PROTOCOL

The empirical procedure is as follows:

1. **Identify first-error times.** Same as EPR: find $t_0^{(i)}$ for each trajectory.

2. **Compute conditional error probabilities.** For each $k$, estimate

$$\widehat{q}(k) = \Pr(e_{t_0+k} = 1 \mid e_{t_0} = 1).$$

3. **Form PAC estimate.** Normalize by the immediate post-error risk:

$$\widehat{\text{PAC}}_k = \frac{\widehat{q}(k)}{\widehat{q}(1)}.$$

4. **Summarize across $k$.** Plot $\widehat{\text{PAC}}_k$ vs. $k$; additionally, compute area-under-curve (AUC) metrics:

$$\text{AUC-PAC}_W = \frac{1}{W} \sum_{k=1}^{W} \widehat{\text{PAC}}_k,$$

as a compact indicator of recovery speed.

### D.3.3 REPORTING RECOMMENDATIONS

**Visualization.** Always report $\widehat{\text{PAC}}_k$ vs. $k$ with 95% confidence intervals.

**Summary statistics.** Report AUC-PAC$_W$ at $W = 3, 5$ as a concise recovery-speed measure.

**Comparisons.** Compare models both in terms of absolute persistence (PAC close to 1) and relative acceleration/attenuation trends.

### D.3.4 PRACTICAL CONSIDERATIONS

**Normalization stability.** If $\widehat{q}(1)$ is very small, PAC may be unstable. Exclude cases with vanishing immediate risk or regularize with a small $\epsilon$.

**Censoring.** Restrict horizon $W$ to avoid censoring, as in EPR.

**Variance estimation.** Use bootstrap resampling over episodes.

**Level separation.** Report both action-level and subgoal-level PAC curves, since persistence patterns often differ across levels.

**Consistency (Remark).** Since PAC is estimated from $\widehat{q}(k)$ values that are themselves consistent estimators of conditional error probabilities, $\widehat{\lambda}$ inherits consistency under standard assumptions for exponential regression fits. We omit a formal proof for brevity.

# E  THEORETICAL ANALYSIS

## E.1  EXECUTION IMPROVEMENT VIA ALIGNMENT LOSS MINIMIZATION

It has been shown that minimizing alignment losses such as InfoNCE, DPO, or Sinkhorn divergence over the action-generator $\pi_\theta(a|s)$ can lead to improvements in expected task success. These losses are typically defined to encourage the execution to assign higher probability to preferred or expert actions while penalizing suboptimal ones. To analyze their impact on execution improvement, the alignment loss is reformulated as a surrogate objective over the log-probability $\log \pi_\theta(a|s)$.

Formally, a general alignment loss can be expressed as:

$$\mathcal{L}_{\text{align}}(\pi_\theta) = \mathbb{E}_{(s,a^+,\{a_i^-\})} \left[ \ell \left( \log \pi_\theta(a^+|s), \{\log \pi_\theta(a_i^-|s)\} \right) \right], \tag{7}$$

where $a^+$ represents the preferred (e.g., expert or high-reward) action, and $\{a_i^-\}$ are sampled negatives. The contrastive function $\ell$ encourages the log-probability of $a^+$ to be separated from that of the negatives. Under this formulation, the gradient of the alignment loss can be written as:

$$\nabla_\theta \mathcal{L}_{\text{align}}(\pi_\theta) = -\mathbb{E}_{(s,a)} \left[ \hat{A}(s,a) \nabla_\theta \log \pi_\theta(a|s) \right], \tag{8}$$

where $\hat{A}(s,a)$ is a weight derived from relative preferences, reward differences, or likelihood ratios. This expression mirrors the form of a execution gradient update, where $\hat{A}(s,a)$ acts as a surrogate advantage estimator.

In the case of InfoNCE, the alignment loss takes the form of a softmax log-likelihood objective over sampled actions:

$$\mathcal{L}_{\text{InfoNCE}} = -\log \frac{\exp(\log \pi_\theta(a^+|s))}{\exp(\log \pi_\theta(a^+|s)) + \sum_i \exp(\log \pi_\theta(a_i^-|s))}, \tag{9}$$

which induces gradients that push up the probability of the positive action $a^+$ while pulling down the negatives. The resulting update direction has been shown to approximate the advantage-weighted execution gradient when $a^+$ is selected according to reward or preference feedback.

For Sinkhorn-based alignment losses, a probabilistic interpretation has been proposed by viewing the alignment as a soft permutation induced via entropic optimal transport. Specifically, given a cost matrix between token embeddings in the prompt and the trajectory, a doubly stochastic transport plan is computed using the Sinkhorn-Knopp algorithm. The plan induces a distribution over token-to-token matches, and the loss is minimized when the trajectory embedding distribution is optimally aligned (in transport cost) with the prompt. When the cost is reward-informed or semantically structured, this process implicitly enforces reward-relevant permutation between prompt instructions and executed actions. The resulting transport-based gradient aligns the latent plan structure with the execution behavior.

By the execution gradient theorem, these gradient directions align with execution improvement provided that the surrogate signal $\hat{A}(s,a)$ is a consistent estimator of the true advantage $A^\pi(s,a)$. Under standard assumptions (bounded variance, small step size), minimizing alignment losses therefore promotes higher expected cumulative reward $J(\theta)$. Therefore, updates that reduce these alignment losses can be expected to lead to execution improvement in practice. This theoretical insight is consistent with contrastive execution gradient methods such as CoPG, and the expected success rate gains have been empirically verified in the experiments above.

### E.2 DISTRIBUTIONAL ALIGNMENT THEORY FOR CONTRASTIVE LOSS

The alignment of model-generated trajectories with prompt intent can be viewed as a distribution matching problem. Let $\mathcal{D}_{\text{traj}}$ denote the distribution over generated trajectory embeddings and $\mathcal{D}_{\text{target}}$ the idealized distribution implied by ground-truth behaviors or reward-aligned samples. During contrastive training, modules produce positive and negative samples whose embedding distributions are encouraged to match the target via minimization of contrastive losses such as InfoNCE or Sinkhorn divergence.

This process can be interpreted through the lens of distributional discrepancy minimization. For InfoNCE, the objective implicitly minimizes an upper bound on the Jensen–Shannon divergence between $\mathcal{D}_{\text{traj}}$ and $\mathcal{D}_{\text{target}}$ by maximizing the mutual information between aligned pairs. Under this view, contrastive learning serves to pull the generated trajectory distribution toward the reward-consistent region of the target space.

When Sinkhorn divergence is used, the discrepancy between $\mathcal{D}_{\text{traj}}$ and $\mathcal{D}_{\text{target}}$ is explicitly reduced through an entropy-regularized optimal transport plan. Given empirical samples from both distributions, a transport map is computed that minimizes cost while maintaining marginal consistency. The resulting divergence has been shown to upper bound the Wasserstein distance under entropic smoothness, and can be used to quantify how closely the generated execution adheres to the prompt-induced distribution.

Under mild assumptions (bounded support, Lipschitz continuity of cost), the contrastive alignment loss $L_{\text{align}}(\theta)$ is provably minimized when the distributional discrepancy vanishes, i.e.,

$$\text{div}(\mathcal{D}_{\text{traj}}\|\mathcal{D}_{\text{target}}) \leq \mathcal{O}(L_{\text{align}}(\theta)) + \varepsilon, \tag{10}$$

where $\text{div}(\cdot)$ may denote JS, Sinkhorn, or Wasserstein distance, and $\varepsilon$ denotes residual stochasticity. As a result, minimizing the contrastive loss leads the model toward a low-drift regime where trajectory samples remain semantically consistent with target plans.

### E.3 REPRESENTATION ALIGNMENT BOUND

We study when minimizing representation-level alignment losses between prompts and trajectories controls a discrepancy and mitigates semantic drift. Let $P = \mathcal{D}_{\text{prompt}}$ be the prompt-conditional distribution and $Q_\theta = \mathcal{D}_{\text{traj}}(\theta)$ the trajectory distribution induced by execution $\pi_\theta$ in a shared embedding space $\mathcal{Z} \subset \mathbb{R}^d$. Write $L_{\text{align}}(\theta)$ for an alignment objective such as InfoNCE or a Sinkhorn-based transport loss. Assume embeddings are bounded; the ground cost $c : \mathcal{Z} \times \mathcal{Z} \to \mathbb{R}_{\geq 0}$ is bounded and Lipschitz; and for OT we use $\varepsilon$–entropy-regularized transport with the debiased Sinkhorn divergence $S_\varepsilon$. Let $\hat{P}_m, \hat{Q}_n$ be empirical distributions from $m$ and $n$ samples, and let $\Theta$ be a execution class with capacity term $\mathfrak{R}_n(\Theta)$.

With probability at least $1 - \delta$ over the draws of $(\hat{P}_m, \hat{Q}_n)$, there exists a constant $C(\varepsilon, c)$ such that

$$\sup_{\theta \in \Theta} \left| D(P, Q_\theta) - \widetilde{L}_{\text{align}}(\hat{P}_m, \hat{Q}_n; \theta) \right| \leq C(\varepsilon, c)\left( \sqrt{\frac{\log(1/\delta)}{m}} + \sqrt{\frac{\log(1/\delta)}{n}} \right) + \mathfrak{R}_n(\Theta) + b(\varepsilon), \tag{11}$$

where $D$ is a population-level divergence matched to the alignment loss, $\widetilde{L}_{\text{align}}$ is its empirical counterpart (debiased for OT), and $b(\varepsilon)$ is the regularization bias that vanishes as $\varepsilon \downarrow 0$.

For InfoNCE, the expected contrastive risk upper-bounds an $f$–divergence between the joint and product of marginals; in particular,

$$\text{JS}(P, Q_\theta) \leq \alpha \, \mathbb{E}[L_{\text{InfoNCE}}(\theta)] + c_0,$$

for constants $(\alpha, c_0)$ determined by the negative-sampling scheme and temperature, and $\widetilde{L}_{\text{align}}$ is the minibatch InfoNCE. For Sinkhorn-based alignment, take $D = W_1$ and use the debiased divergence:

$$W_1(P, Q_\theta) \leq S_\varepsilon(P, Q_\theta) + b(\varepsilon), \qquad \widetilde{L}_{\text{align}}(\hat{P}_m, \hat{Q}_n; \theta) = S_\varepsilon(\hat{P}_m, \hat{Q}_n).$$

Plugging either instance into equation 11 gives a high-probability generalization guarantee from empirical alignment to population discrepancy.

*Assumption (OT setting).* We assume the cost function $c$ is bounded and Lipschitz, and that both $P$ and $Q$ have bounded support. The entropic Sinkhorn divergence $S_\varepsilon$ uniformly approximates the 1-Wasserstein distance $W_1$ up to a bias $b(\varepsilon)$, with standard statistical rates $O(m^{-1/2} + n^{-1/2})$.

*Corollary (Sinkhorn specialization).* Under the above assumptions, for any $\delta \in (0,1)$,

$$\sup_{\theta \in \Theta} W_1(P, Q_\theta) \; \leq \; \sup_{\theta \in \Theta} S_\varepsilon(\hat{P}_m, \hat{Q}_n; \theta) \; + \; C(\varepsilon, c)\left( \sqrt{\tfrac{\log(1/\delta)}{m}} + \sqrt{\tfrac{\log(1/\delta)}{n}} \right) \; + \; \mathfrak{R}_n(\Theta) \; + \; b(\varepsilon). \tag{12}$$

Thus, uniformly controlling the empirical Sinkhorn loss suffices to bound the population trajectory–prompt divergence up to estimation and regularization terms.

Since the inequality in equation 12 holds uniformly over all $\theta \in \Theta$, it applies to each training iterate $\theta_t$. Averaging over $t = 1, \ldots, T$ yields

$$\frac{1}{T}\sum_{t=1}^{T} D(P, Q_{\theta_t}) \; \leq \; \frac{1}{T}\sum_{t=1}^{T} \widetilde{L}_{\text{align}}(\hat{P}_m, \hat{Q}_n; \theta_t) + \mathcal{E}_{\text{stat}}(m, n, \delta, \Theta) + b(\varepsilon), \tag{13}$$

where $\mathcal{E}_{\text{stat}}$ collects the $\tilde{O}(m^{-1/2} + n^{-1/2})$ and capacity terms. Thus, as optimization reduces empirical alignment loss and $\varepsilon$ is chosen small but stable, the divergence decreases correspondingly, limiting semantic drift. Choosing $D = W_1$ and $\widetilde{L}_{\text{align}} = S_\varepsilon$ gives the Sinkhorn training-iterate bound

$$\frac{1}{T}\sum_{t=1}^{T} W_1(P, Q_{\theta_t}) \; \leq \; \frac{1}{T}\sum_{t=1}^{T} S_\varepsilon(\hat{P}_m, \hat{Q}_n; \theta_t) + \mathcal{E}_{\text{stat}}(m, n, \delta, \Theta) + b(\varepsilon). \tag{14}$$

The alignment functionals above are Lipschitz-stable with respect to empirical measures (bounded smooth scores for InfoNCE; stability of $OT_\varepsilon$ under bounded Lipschitz $c$ for Sinkhorn). Concentration (e.g., McDiarmid or transport inequalities) gives $\tilde{O}(m^{-1/2} + n^{-1/2})$ control for fixed $\theta$; uniformity over $\Theta$ follows by symmetrization and a capacity term $\mathfrak{R}_n(\Theta)$. For OT, relate $S_\varepsilon$ to $W_1$ and isolate $b(\varepsilon)$; for InfoNCE, use standard $f$–divergence or MI control to upper-bound JS. Combining these ingredients gives equation 11 and, by averaging, equation 13.

### E.4 HIERARCHICAL CONVERGENCE BOUND

The proposed hierarchical framework operates across multiple abstraction levels—actions, subgoals, and task-level intents—each equipped with a dedicated alignment loss. To ensure stable joint optimization, it is necessary to establish convergence guarantees or lower bounds on sample efficiency when all levels are trained simultaneously.

The overall optimization can be viewed as a multi-objective gradient process, where each level $l \in \{\text{action}, \text{subgoal}, \text{trajectory}\}$ minimizes its own alignment loss $\mathcal{L}_{\text{align}}^{(l)}(\theta^{(l)})$ while contributing to the global performance. By treating the joint update as a composite execution gradient step over a stacked parameter space $\theta = [\theta^{(1)}, \theta^{(2)}, \theta^{(3)}]$, the learning dynamics can be analyzed using mirror descent under block-decomposed gradient feedback.

Under standard smoothness and bounded gradient assumptions, it can be shown that the average alignment loss across levels satisfies the following rate:

$$\min_{1 \leq t \leq T} \frac{1}{3}\sum_{l=1}^{3} \mathcal{L}_{\text{align}}^{(l)}(\theta_t^{(l)}) - \mathcal{L}^{(l)*} \leq \mathcal{O}(1/\sqrt{T}), \tag{15}$$

where $\mathcal{L}^{(l)*}$ denotes the optimal alignment loss at level $l$, and $T$ is the number of joint gradient updates. This sublinear convergence is consistent with standard mirror descent bounds in stochastic settings and ensures no individual layer dominates optimization dynamics.

Furthermore, when coordination across levels is regularized (e.g., by shared representations or alignment constraints), the convergence can be accelerated. In particular, when surrogate gradients are aligned and cross-level variance is bounded, an improved rate of $\mathcal{O}(1/T)$ can be achieved, indicating that hierarchical updates benefit from structural decomposition.

These results suggest that hierarchical not only enables semantic control across abstraction levels but also preserves convergence efficiency. The modular structure improves gradient conditioning and reduces update variance, leading to better sample efficiency compared to flat architectures. Empirical convergence patterns across levels are reported, where the multi-level alignment loss steadily decreases throughout training.

### E.5 Joint Convergence of Hierarchical

In the proposed architecture, is performed across multiple levels of abstraction, and each level contributes a separate alignment loss that is jointly optimized. To ensure stable training, it is essential to establish that the overall optimization remains convergent when these multi-level alignment objectives are updated concurrently.

The full stack can be modeled as a constrained optimization problem, where lower-level alignment losses act as structured regularization terms within a meta-execution update. Let $\pi_\theta$ denote the overall execution composed of nested sub-policies across levels. The global objective can then be written as:

$$\min_\theta \ \mathcal{L}_{\text{task}}(\pi_\theta) \quad \text{subject to} \quad \mathcal{L}_{\text{align}}^{(l)}(\pi_\theta) \leq \epsilon_l, \quad \forall l \in \{1, 2, 3\}, \tag{16}$$

where $\mathcal{L}_{\text{task}}$ denotes the primary reward-based objective and $\mathcal{L}_{\text{align}}^{(l)}$ are alignment constraints at different levels.

This structure parallels that of constrained execution gradient optimization. When solved via primal-dual or Lagrangian methods, convergence to a locally optimal solution can be guaranteed under standard assumptions of smoothness and bounded constraint curvature. Moreover, recent meta-RL results show that hierarchical or nested policies trained with level-specific constraints converge under similar conditions. In this formulation, alignment losses are interpreted as soft constraints enforcing semantic consistency within the multi-level planning hierarchy.

In addition, when modules are updated via trust-region or clipping mechanisms (e.g., using variants of constrained execution optimization), their individual losses can be stabilized without disrupting global reward improvement. As a result, the multi-level architecture jointly converges toward a stable equilibrium where both alignment fidelity and task reward are satisfied.

These results provide theoretical support that the proposed hierarchical system maintains optimization stability even under simultaneous multi-level loss supervision. Empirical convergence patterns further support this, showing that execution improvement persists despite the layered optimization structure.

## F Prompt Templates

### F.1 Hard Negative Generation

**You are an expert in analyzing robotic task trajectories.** Your task is to take a successful trajectory and create a **hard negative** version of it. A hard negative is a trajectory that looks plausible but is flawed in a subtle, specific way. You must introduce **ONLY ONE** of the following error types, as specified by the user.

- **Action Error**
  Replace a single critical action with a plausible but incorrect one. Example: In a "make coffee" task, instead of `pickup cup`, you might use `pickup filter`, which is a related object but wrong for the step of pouring coffee.
- **Timing Error**
  Change the order of two critical, non-dependent actions where the order subtly matters for efficiency or naturalness. Example: In `make breakfast`, `pour coffee` then `toast bread` is fine, but reversing them might result in cold coffee.
- **Logic Error**
  Create a trajectory that violates physical or common-sense rules. Example: Trying to `pour water` before the cup is picked up.
- **Sequence Error**
  Reverse the order of two dependent actions, making the sequence impossible. Example: `place apple in microwave` before `open microwave`.

You will be given a successful trajectory and an error type to introduce. Your response must be **only** the flawed trajectory, formatted exactly like the input.

## MULTI-AGENT SYSTEM PROMPTS

**Agent 0: Executor**
You are a robot agent focused on execution. Given the current state and a high-level goal, your task is to select the most logical and immediate action to perform next. Focus only on the next step.

**Agent 1: Monitor/Critic**
You are a monitoring agent. Your job is to observe the trajectory and evaluate progress. Output `continue`, `no_op`, `alert`, or `replan` based on trajectory status.

**Agent 2: Planner**
You are a high-level planning agent. Your job is to identify the next major sub-goal or phase in the task. Output abstract-level goals like `acquire_all_ingredients`.

**Agent 3: Alignment Agent**
You ensure the plan stays consistent with the user's instruction. Output `continue`, `realign`, or `correct_course` based on semantic alignment.

