# OpenReview forum: "ReCAPA: Hierarchical Predictive Correction to Mitigate Cascading Failures"
_ICLR.cc/2026/Conference — ICLR 2026 Poster_

### Official Review · Reviewer_3npG · 2025-10-31

**Soundness:** 2
**Presentation:** 1
**Contribution:** 2
**Rating:** 4
**Confidence:** 3

**Summary:**

The paper proposes ReCAPA (Reflective Contrastive Alignment and Planning Architecture), a hierarchical predictive correction framework designed to mitigate cascading failures in long-horizon reasoning for vision–language–action (VLA) agents. Unlike prior methods that rely on fixed task decomposition or post-hoc correction, ReCAPA proactively anticipates and corrects deviations across three hierarchical levels—actions, subgoals, and trajectories—using predictive contrastive learning and prompt–trajectory alignment modules based on Sinkhorn optimal transport and score-field gradients. The authors also introduce two diagnostic metrics, Error Propagation Rate (EPR) and Propagation Attenuation Coefficient (PAC), to evaluate how errors accumulate and dissipate during execution. Experiments on VisualAgentBench, MineDojo, and MAP-THOR demonstrate that ReCAPA achieves higher success rates and superior robustness compared to strong LLM baselines, effectively reducing error propagation in multi-step embodied tasks.

**Strengths:**

The paper tackles an important and underexplored issue—cascading failures in long-horizon reasoning—and proposes a technically sound hierarchical correction framework. The introduction of predictive alignment across multiple levels and the new diagnostic metrics (EPR and PAC) provide useful analytical tools for evaluating robustness in embodied agents. Experiments are comprehensive, spanning several major benchmarks, and the reported improvements are consistent across tasks.

**Weaknesses:**

1. Conceptual novelty is moderate. The proposed hierarchical predictive correction (HPCC) framework largely builds upon existing ideas in self-reflective planning (e.g., Reflexion, ReAct, AdaPlanner) and hierarchical alignment (e.g., HiP, TrajPrompt). The distinction between ReCAPA’s “predictive correction” and prior feedback-based reflection mechanisms is not sharply articulated. The paper would benefit from a clearer theoretical or algorithmic differentiation beyond combining multi-level prediction and Sinkhorn-based alignment.

2. Many components—such as how cross-level corrective gradients interact with the execution network or how the LLM’s decomposition is integrated during inference—are described only qualitatively. Key design choices (e.g., the dimensionality of embeddings, predictor architectures, and training stability) are omitted, making the method hard to reproduce or verify.

3. The paper emphasizes “hierarchical correction,” yet no qualitative rollout visualizations or case studies are shown to illustrate how errors are detected and corrected in practice. Examples demonstrating the evolution of alignment or prediction across levels would make the mechanism more convincing.

4. The new EPR and PAC metrics are interesting but lack correlation studies with task performance or user interpretability. It remains unclear whether improvements in these metrics genuinely indicate better planning robustness or simply reflect model-specific biases.

5. Writing and organization issues. The text is dense and occasionally repetitive, with unclear boundaries between related work and method sections. Some equations (e.g., Eq. 1–4) are introduced abruptly without sufficient intuition. The overall presentation would benefit from clearer motivation and more concise explanation of the technical pipeline.

**Questions:**

The proposed EPR and PAC metrics are intriguing, but how sensitive are they to task complexity or trajectory length? Have the authors verified that improvements in these metrics correlate with human-judged robustness or actual task success rates?

I also notice some typo of writing:

1. The appendix title: “Appendix C.3 ABLATION ON PREDICTION AND” seems incomplete.

2. The first paragraph of Related Work has repeat sentence.

---

> ### Author Response · Authors · 2025-11-26
> **Response to Reviewer 3npG**
>
> We sincerely appreciate your evaluation of our work and your questions. Below, we address each point with clarifications and additional explanations.
>
> ##
>
> **Q1:** What is the key difference between ReCAPA and existing methods ?
>
> **A1:**
>
> 1. **Planning**：ReCAPA’s goal is to improve **planning** by performing predictive correction during execution. It does **not rely on external reflection signals**, which are the core of reflection methods such as Reflexion [1], ReAct [2], and AdaPlanner [3].
> 2. **Cross-level correction**：HiP [4] only decomposes tasks without enforcing subgoal–action alignment, while TrajPrompt [5] aligns only at the trajectory level and cannot correct fine-grained action drift. In contrast, ReCAPA performs **cross-level** semantic prediction and correction, keeping lower-level representations continually **aligned** with higher-level intent.
>
>  ##
>
> **Q2:** How cross-level corrective gradients interact with the execution network?
>
> **A2:**
>
> 1. **Supervised gradients**：The execution network is trained with ground-truth actions from the dataset, and these supervised loss update the shared action-level encoder.
>
> 2. **Cross-level corrective gradients**：The cross-level prediction and alignment losses send corrective gradients into all three hierarchical encoders, and the execution network relies on the representations **produced by these encoders** for **action selection**.
>
>    As a result, the execution network and the hierarchical predictors are jointly optimized through shared encoder parameters.
>
> ##
>
> **Q3:** How the LLM’s decomposition is integrated during inference?
>
> **A3:** The LLM provides the full task **decomposition into subgoals**, and ReCAPA then checks each action against these subgoal embeddings, correcting mismatch during execution.
>
> ##
>
> **Q4:** Embedding dimensions and training stability are not provided.
>
> **A4:** The embeddings are set to 512 (low-level), 1024 (mid-level), 2048 (high-level).
>
>  All layers first project to hidden_dim=512 → Transformer → final projection.
>
>  Predictor uses MLP + Linear + LayerNorm + ReLU + Dropout.
>
>  Training stability: Adam, warmup LR, Dropout 0.1, LayerNorm.
>
> ##
>
> **Q5:** Show some qualitative rollout visualizations or case studies to illustrate “hierarchical correction”.
>
> **A5:** We provide a step-by-step illustration of how ReCAPA detects and corrects errors during inference **and a Figure in Page 22 in PDF**.
>
> Example task: “Take the pot from the left burner, wash it, and turn on the burner.”
>
> Step 1: Correct execution begins.
>
> `[WALK]_(burner)`  →  `[MOVE]_(pot)`   →  `[TURN_ON]_(burner)`    →    `[WALK]_(sink)`
>
> Step 2: Action-level error
>
>   `[PUT]_(pot)`   ← incorrect action
>
> The action-level representation fails to predict the correct subgoal-level semantic target “  put the pot into the sink ”, causing action–subgoal alignment to drop. Subgoal-level forecasting then drifts toward an incorrect direction, which subsequently causes global prompt–trajectory alignment to decrease. ReCAPA identifies this as a deviation.
>
> Step 3: Action-level correction **(resample)**
>
> Resample until the pot is placed in the sink.
>
> Step 4: Execution resumes and completes
>
> `[WALK]_(burner)`  →  `[MOVE]_(pot)`   →  `[TURN_ON]_(burner)`    →    `[WALK]_(sink)` →     `[PUT]_(pot)`   → `[Turn_on]_(tap)`
>
>
>
>
> ##
>
> References:
>
> [1] Shinn, N., Cassano, F., Gopinath, A., Narasimhan, K., & Yao, S. (2023). Reflexion: Language agents with verbal reinforcement learning. *Advances in Neural Information Processing Systems*, *36*, 8634-8652.
>
> [2] Yao, S., Zhao, J., Yu, D., Du, N., Shafran, I., Narasimhan, K. R., & Cao, Y. (2022, October). React: Synergizing reasoning and acting in language models. In *The eleventh international conference on learning representations*.
>
> [3] Sun, H., Zhuang, Y., Kong, L., Dai, B., & Zhang, C. (2023). Adaplanner: Adaptive planning from feedback with language models. *Advances in neural information processing systems*, *36*, 58202-58245.
>
> [4] Ajay, A., Han, S., Du, Y., Li, S., Gupta, A., Jaakkola, T., ... & Agrawal, P. (2023). Compositional foundation models for hierarchical planning. *Advances in Neural Information Processing Systems*, *36*, 22304-22325.
>
> [5] Tsao, L. W., Tsui, H. T., Tuan, Y. R., Chen, P. C., Wang, K. L., Wu, J. C., ... & Cheng, W. H. (2024, September). TrajPrompt: Aligning color trajectory with vision-language representations. In *European Conference on Computer Vision* (pp. 275-292). Cham: Springer Nature Switzerland.

---

> > ### Author Response · Authors · 2025-11-26
> > **The rest of response to Reviewer 3npG**
> >
> > **Q6**:  The EPR and PAC metrics lack correlation studies.
> >
> > **A6:** Although EPR and PAC are not theoretically required to correlate with task success, in practice more stable planning (low EPR and stable PAC) consistently leads to higher success rates. Below we additionally report the correlation between EPR/PAC and MAP-THOR SR.
> >
> > | Metric Pair | Spearman $ρ$ | Pearson $ r$ |
> > | ----------- | -------------- | -------------- |
> > | PAC vs SR   | 0.9            | 0.98           |
> > | EPR vs SR   | −0.80          | −0.94          |
> >
> > This is the table of correlation between EPR/PAC and SR on VisualAgentBench.
> >
> > | Metric Pair | Spearman $ρ$ | Pearson $ r$ |
> > | ----------- | :------------- | -------------- |
> > | PAC vs SR   | 0.9            | 0.93           |
> > | EPR vs SR   | −0.90          | −0.96          |
> >
> > Our multi-seed results show a remarkably consistent trend: models with **lower EPR and more stable PAC** reliably achieve **higher SR**. When a model both corrects local errors efficiently (high PAC) and prevents error accumulation over time (low EPR), its trajectories remain close to semantically valid plans, which naturally yields higher success rates even under stochastic environments and different random seeds.
> >
> > ##
> >
> > **Q7:** Some equations (e.g., Eq. 1–4) are introduced abruptly without sufficient intuition.
> >
> > **A7:**  In the PDF we provide additional clarification on the introduction of some equations **written in Blue.**
> >
> > ##
> >
> > **Q8:** How sensitive are EPR and PAC to task complexity or trajectory length
> >
> > **A8:** EPR and PAC naturally vary with trajectory length, but our results show that their relative ordering across models remains stable.  PAC fluctuates more on short horizons because a single mistake has a larger proportional effect, while both metrics become more stable as horizons grow.
> >
> > ##
> >
> > **Q9:** The appendix title: “Appendix C.3 ABLATION ON PREDICTION AND” seems incomplete.
> >
> > **A9:** We revise it to “Appendix C.3 Ablation on Prediction and Alignment.” now.
> >
> > ##
> >
> > **Q10:** The first paragraph of Related Work has repeat sentence.
> >
> > **A10:** We revise it now. Please look at the newly uploaded PDF.

---

> > > ### Comment · Reviewer_3npG · 2025-11-26
> > >
> > > Thank you for addressing my concerns. I appreciate the clarifications provided, as well as the updates made in the new PDF, which have helped resolve some of my concerns. It is clear that effort was put into improving the presentation.
> > >
> > > However, I still believe that the contributions and presentation of the paper have certain limitations. Given other reviewer's comment, I have decided to raise my score to 6.

---

> > > > ### Author Response · Authors · 2025-11-27
> > > >
> > > > Thank you very much for taking the time to review our responses.  Your positive feedback and support serve as an encouragement to our research efforts.

---

### Official Review · Reviewer_rzHw · 2025-10-31

**Soundness:** 3
**Presentation:** 3
**Contribution:** 3
**Rating:** 8
**Confidence:** 3

**Summary:**

This paper introduces ReCAPA, a predictive correction framework designed to mitigate cascading failures in multi-step reasoning in the context of VLA models. The main ides is that small errors in subgoal or action specification can compound over time, degrading overall performance.
ReCAPA addresses this by applying predictive correction mechanisms at multiple levels such as actions, subgoals, and trajectories with the aim of anticipating and correcting deviations before they propagate.
The proposed method demonstrates strong performance, reportedly outperforming both proprietary and open-source large language models on benchmark tasks.

**Strengths:**

- The paper is clear, well-structured, and easy to follow, with an interesting and well-motivated idea.
- The method is explained clearly, with a logical progression from motivation to implementation.

**Weaknesses:**

- The discussion of limitations lack details. While two limitations are mentioned, the proposed mitigation strategies are not empirically validated, which reads as somewhat unbalanced.
- The statistical significance of results is unclear. Although the authors mention using three random seeds on VisualAgentBench (Fig. 4), it is not evident how consistent or significant the improvements are across other experiments.
- The reproducibility of the results is limited: the authors do not provide the code, and the training details are not discussed.
- The ablation studies are limited.

**Questions:**

- Why did the authors not further investigate the proposed mitigation strategies for ReCAPA’s limitations?
- Could the authors include or discuss ablation analyses to clarify the contribution of each module?
- Can the authors elaborate on the statistical significance of their results beyond the limited VisualAgentBench trials?

---

> ### Author Response · Authors · 2025-11-26
> **Response to Reviewer rzHw**
>
> We sincerely appreciate your positive evaluation of our work and your questions. We are grateful for the time you devoted to reviewing our submission. Below, we address each point with clarifications and additional explanations.
>
> ##
>
> **Q1:** The limitations section lacks depth, as the proposed mitigations are not empirically supported.
>
> **A1:** Our intention in the limitations section is simply to acknowledge the current boundaries of ReCAPA rather than to claim additional methodological components. The strategies mentioned are presented only as potential future directions, and we did not experiment with them because they require substantial design and validation beyond the scope of this work. We revise the section to make this distinction clearer in color **Blue**. **Please see the PDF**.
>
> ##
>
> **Q2:** Show statistical significance of results on all benchmarks.
>
> **A2:** Thank you for pointing this out. All experiments on VisualAgentBench, MineDojo, and MAP-THOR were conducted using three independent random seeds.
>
> **MAP_THOR:**
>
> | Metric   | Seed 1 | Seed 2 | Seed 3 | Average  |
> | -------- | ------ | ------ | ------ | -------- |
> | SR       | 0.74   | 0.73   | 0.77   | **0.75** |
> | TR       | 0.92   | 0.93   | 0.95   | 0.93     |
> | Coverage | 0.95   | 0.98   | 0.93   | 0.95     |
> | Balance  | 0.93   | 0.92   | 0.94   | 0.93     |
>
> **MineDojo:**
>
>  **Zero-shot 3-seed**
>
> | Task          | Seed 1 | Seed 2 | Seed 3 | Average |
> | ------------- | ------ | ------ | ------ | ------- |
> | stick         | 0.53   | 0.6    | 0.57   | 0.57    |
> | craftingtable | 0.4    | 0.4    | 0.5    | 0.43    |
> | woodenpickaxe | 0.67   | 0.67   | 0.67   | 0.67    |
> | furnace       | 0.57   | 0.67   | 0.57   | 0.6     |
> | milkbucket    | 0.83   | 0.87   | 0.79   | 0.83    |
> | wool          | 0.7    | 0.61   | 0.7    | 0.67    |
> | beef          | 0.5    | 0.63   | 0.46   | 0.53    |
> | mutton        | 0.38   | 0.33   | 0.35   | 0.35    |
> | bed           | 0.3    | 0.3    | 0.3    | 0.3     |
>
> **Iter-3 3-seed**
>
> | Task          | Seed 1 | Seed 2 | Seed 3 | Average |
> | ------------- | ------ | ------ | ------ | ------- |
> | stick         | 0.65   | 0.6    | 0.62   | 0.62    |
> | craftingtable | 0.62   | 0.67   | 0.67   | 0.65    |
> | woodenpickaxe | 0.85   | 0.8    | 0.75   | 0.80    |
> | furnace       | 0.67   | 0.82   | 0.7    | 0.73    |
> | milkbucket    | 0.93   | 0.96   | 0.96   | 0.95    |
> | wool          | 0.7    | 0.7    | 0.78   | 0.73    |
> | beef          | 0.56   | 0.6    | 0.63   | 0.6     |
> | mutton        | 0.59   | 0.5    | 0.5    | 0.53    |
> | bed           | 0.4    | 0.45   | 0.32   | 0.4     |
>
> Across all benchmarks, the three-seed results demonstrate that ReCAPA’s performance is **stable** and **not driven by single-run variance**.
>
> ##
>
> **Q3:** The paper provides neither code nor enough training details.
>
> **A3:**  Our code is in **Supplementary Materials**.
>
> Our appendix includes key hyperparameters such as learning rate, batch size, optimizer choice, hierarchical loss weights, gating thresholds, and rollout configurations. In the future version, we will include more details: (1) the full prompting templates used during correction and trajectory generation, (2) the procedure for generating negative state–action sequences via GPT-4o-mini, and (3) a clearer description of the overall training protocol.
>
> ##
>
> **Q4:** The ablation studies are limited.
>
> **A4:** Due to strict space constraints in the main paper, we only included the most essential ablation study. The remaining analyses, **including layer-wise ablation, coupling-strategy comparisons, prediction-versus-correction effects, and gradient-based execution-update studies,** are provided in detail in Appendix C.
>
> ##
>
> **Q5:** Why not further investigate the proposed mitigation strategies for limitations?
>
> **A5:** The mitigation strategies mentioned in the limitations section were intended as forward-looking directions rather than components of ReCAPA. We did not investigate them experimentally because each requires substantial new module design, additional objectives, and extensive validation, which would constitute a separate line of research beyond the scope of this work.

---

> > ### Comment · Reviewer_rzHw · 2025-11-27
> > **Reply**
> >
> > Thank you for your detailed responses to my questions. I appreciate your efforts in clarifying the points that were causing confusion for me. I now have a much better understanding of most of the aspects that were unclear.
> >
> > I am hoping to see the additional details discussed above in the revised version of the paper and continue to support acceptance of the paper.

---

> > > ### Author Response · Authors · 2025-11-29
> > >
> > > We are deeply grateful for your careful review and valuable feedback. We will add the content mentioned by rebuttal in the further version.

---

### Official Review · Reviewer_iUSv · 2025-11-01

**Soundness:** 3
**Presentation:** 3
**Contribution:** 4
**Rating:** 8
**Confidence:** 3

**Summary:**

The paper focuses on the problem of proactive error prediction and corrections for VLA models. The method, Reflective Contractive Alignment and Planning Architecture (ReCAPA) incorporates a bottom-up approach for deviation detection, and a top-down approach for deviation correction. ReCAPA aims to tackle the challenges of both short-term action error correction as well as long-term plan adherence. The hierarchical framework breaks down an embodied task trajectory into action, subgoal, and trajectory levels. At high level, the model aims to achieve prompt-trajectory distributional alignment through the Sinkhorn-based module. At the action and subgoal level, the model aims to achieve fine-grained prediction and execution alignment through the Score-fieldSong module. The model goes through 2 training stages: a pre-training to align state-action sequences, and a joint training of hierarchical model through a combined loss of Optimal Transport between the task prompt embedding and the overall trajectory distribution, a denoting local objective to pull fine-grained actions and subgoals closer to the prompt semantic intent, and a contrastive corrective loss between each adjacent levels (e.g.. action-subgoal, subgoal-trajectory). The paper also proposed two new evaluation methods, error propagation rage (EPR), and propagation attenuation coefficient (PAC) to explicitly measure the error propagation and cascading degree during the long horizon process. The paper conducted experiments on 3 benchmarks, and compared aganist numerous open sourced and proprietary LLMs, and demonstrated improved overall success rate as well as some EPR and PAC curves. The paper also conducted ablation studies to evaluate the importance of each component.

**Strengths:**

- The paper is well motivated and well structured. It studies the problem of action deviation from both short term mistake correction and longer term task alignment two objectives.
- The paper proposed an interesting method to decompose the trajectories into hierarchical 3 levels, and align the predicted states with prompts for deviation detection and correction.
- The paper offers sufficient and concise explanations behind the method, loss function designs, and acknowledge the key limitations in the conclusion section while offering future directions.
- The paper conducted thorough experimentations against multiple benchmarks and baseline models, offering sufficient evidence to support the advantages of the proposed method.

**Weaknesses:**

1. Table 1: it would be helpful to explain how 'transport rate, coverage, and balance' these three metrics are calculated
2. Questions below

**Questions:**

1. When an inconsistencies arises at state $S_i$, with action $a_i$, ($S_i \rightarrow a_i$), and the model learns and updates its weight, does it learn a 'correction action' from the current state ($S_i \rightarrow a_i \rightarrow a_j \rightarrow S_j$) or a hopefully better action $a_j$ from the same state $S_i$, ($S_i \rightarrow a_j$)?

2. Section 3.3.2: the prompt embeddings $p$ -- are these the same prompts as in Section 3.3.1 $v$ for the overall task, or are they action/substep specific prompts?

3. Ln 257: how to generate negative state-action sequences from GPT-4o-mini?

4. If a model performs consistently well throughout a task, will the PAC score be high or low?
3. Ln 114-115: duplicates

---

> ### Author Response · Authors · 2025-11-26
> **Response to Reviewer iUSv**
>
> We sincerely appreciate your positive evaluation of our work and your questions. We are grateful for the time you devoted to reviewing our submission. Below, we address each point with clarifications and additional explanations.
>
> ##
>
> **Q1:** Explain 'transport rate, coverage, and balance'
>
> **A1:** These three indicators come from the MAP-THOR [1].
>
>  We follow MAP-THOR for the evaluation metrics and present the detailed explanations below:
>
> - **Transport Rate (TR)** calculates the percentage of transport-related subtasks that the agent completes successfully.
> - **Coverage** calculates the proportion of all interactions in the environment that the agent executes successfully.
> - **Balance** measures how evenly the agent’s successful actions are distributed across all subtasks.
>
> We will include these definitions in the future version for clarity.
>
> ##
>
> **Q2:** Is the model learning a corrective chain of actions, or just a better next action from the same state?
>
> **A2:** ReCAPA follows the version where the model learns to **choose a better action $a_j$** directly from the same state $S_i$. When an incorrect action occurs, Hierarchical Predictive Contrastive Correction (HPCC) adjusts the lower-level representation so that, upon encountering the same state again, the model directly selects a better next action.
>
> ##
>
> **Q3:** Is $p$ the same global task prompt as $v$, or an action-specific prompt?
>
> **A3:** The prompt embedding $p$ in Section 3.3.2 is the same as in Section 3.3.1, and we do not generate action-specific or substep-specific prompts.
>
> ##
>
> **Q4:** How to generate negative state-action sequences from GPT-4o-mini?
>
> **A4:** We generate negative samples by prompting GPT-4o-mini to produce perturbed variants of the original action sequence. These perturbations include **deleting individual steps, reordering the sequence, and replacing certain actions** with semantically related but incorrect alternatives.
>
> ##
>
> **Q5:** If a model performs consistently well throughout a task, will the PAC score be high or low?
>
> **A5:** A consistently strong model will exhibit a **high PAC score**, because its recovery from small deviations is fast and the risk dissipates quickly along the steps.
>
> ##
>
> **Q6:** Ln 114–115: duplicates
>
> **A6:** Thank you for pointing this out. We revise it now. Please look at the newly uploaded PDF.
>
>
>
> ##
>
> **Reference**:
>
> [1] Nayak, S., Orozco, A. M., Have, M. T., Thirumalai, V., Zhang, J., Chen, D., ... & Balakrishnan, H. (2024). *LLaMAR: Long-Horizon Planning for Multi-Agent Robots in Partially Observable Environments*. arXiv preprint arXiv:2407.10031.

---

### Meta-Review · Area_Chair_6NQC · 2026-01-04

**Summary:**

**Summary.** The paper studies an interesting and important question in VLA: how to prevent cascade failures in long-horizon tasks. The authors proposed HPCC and execution modules to predict outcomes and proactively improve actions in a hierarchical manner.  The authors further proposed two evaluation metrics and conducted extensive experiments.

**Strength (by reviewers).** Well motivated, structured, and explained paper; important problem to address; interesting and sound method; useful new metrics; thorogh experiments

**Weakness (by reviewers).** Insufficient implementation details and design choice justification; insufficient background information; missing statistical significance; insufficient ablation studies; questionable reproducibility (no code); moderate novelty; no quantitative examples; writing and organization can be improved.

**Decision.** The paper received an average score of 6.67 (4, 8, 8) in the beginning. The authors provided rebuttals, and two reviewers responded the the rebuttals, indicating (partial) satisfaction. Specifically, Reviewer 3npG, who initially gave 4, indicated the intent to raise the score to 6. The AC read the paper, reviews, rebuttals, and discussions. The AC thinks that the rebuttals have addressed many questions/concerns. Given that the paper has clear strengths and Reviewer 3npG intended to increase the score to 6 (so all three reviewers are leaning towards acceptance), the AC recommends acceptance.

**AC's further suggestion.** That said, the AC does agree with Reviewer 3npG's remaining concerns and the issues underlying other reviewers' original concerns/questions. That is, the current paper writing (main paper) stays at a high level, making it hard to understand 1) the design choices, 2) the training and inference processes, 3) how the whole framework actually works (e.g., examples). While the authors mentioned that many details are in the Suppl., the AC suggests that the authors do a better job in improving the main paper to incorporate these missing contents. Please note that supplementary materials (including the appendix text) are not necessarily to be read by the reviewers (and future readers). The AC suggests that the authors think from a reader's perspective---how to get a clear picture of the framework, design choice, and sketch of the implementation details in 9 pages. References to the supplementary materials should be given in the main paper. Pseudocode is also suggested.

Further, the authors should have done a better job in addressing Reviewer 3npG's concern about *Conceptual novelty is moderate.* Instead of responding with a few sentences, the AC suggests that the authors provide a detailed discussion section in the camera-ready version.

**Reviewer Concerns:**

**Reviewer iUSv:**
- **Addressed:** Metric definition; better vs. correct actions; other clarification and implementation detail questions
- **Remaining:** None

**Reviewer rzHw:**
- **Addressed:** About limitation and mitigation; statistical significance; reproducibility & implementation details; ablation studies
- **Remaining:** None

**Reviewer 3npG:**
- **Addressed:** Improved writing and organization, examples, implementation details, discussion on new metrics
- **Remaining:** Better writing and organization; novelty and contributions (partially addressed)

**Reviewer Scores:**

- **Reviewer iUSv (8 to 8):** The reviewer will likely keep the score. The score was already high; the authors' rebuttal addressed the questions.

- **Reviewer rzHw (8 to 8):** The reviewer will likely keep the score. The score was already high; the reviewer responded to the rebuttal on 27 Nov 2025, 09:59, acknowledging that it helped the reviewer better understand the paper and resolve confusion. The reviewer indicated support for paper acceptance.

- **Reviewer 3npG (4 to 6):** On 26 Nov 2025, 04:26, the reviewer responded to the rebuttal, indicating the intention to **increase the score to 6**. Specifically, the reviewer acknowledged that some of the concerns have been addressed, especially in improving the presentation. Still, some concerns remained not fully addressed.

---

### Decision · Program_Chairs · 2026-01-26

Accept (Poster)